# BinSGDM: Extreme One-bit Quantization for Communication Efficient Large-Scale Distributed Training

## Abstract

To alleviate the communication bottleneck of large-scale distributed training, a rich body of prior communication-compression optimizers have been proposed. These methods focus mainly on a high compression ratio to expect acceleration. However, some recent works pointed out, when running with distributed training frameworks ( *e.g.*, *DistributedDataParallel* in Pytorch), these methods provide no acceleration over the off-the-shelve uncompressed *SGD/Adam* in the typical settings, due to heavy compression/decompression computation or incompatibility with efficient communication primitives or the requirement of uncompressed warmup at the early stage. For these reasons, we propose a novel extreme one-bit quantization optimizer, dubbed *BinSGDM*. The quantization of *BinSGDM* is computed easily and lightly, and it does not need to resort to uncompressed optimizers for warmup. We also theoretically prove that it promises the same convergence speed as the original *Adam*. Moreover, we specially present a hierarchical 1-bit *All-Reduce* technique to further lower the communication volume. Extensive experiments are conducted on 8 to 64 GPUs (1 to 8 nodes) for distributed training with *DistributedDataParallel*, and the experimental results demonstrate that *BinSGDM* with the communication scheme can achieve up to $\mathbf{2.5\times}$ speedup for training ResNet-50 and $\mathbf{6.3\times}$ speedup for training BERT-Base, compared to the full-precision optimizers.

## 1 Introduction

With the rapid development of computational power, "bigger" and "bigger" deep neural network (DNN) models are proposed for expecting better performance, from the early classical models, such as AlexNet(61M parameters) (Krizhevsky et al. (2017)), and ResNet (ResNet-50: 20.5M parameters) (He et al. (2016)) to the current foundation models, such as BERT (BERT-Lagre: 340M parameters )(Devlin et al. (2018)), and GPT (GPT-3: 176B parameters)(Brown et al. (2020)). Scalable parallelism across distributed computing workers for training these large-scale models becomes a necessity. During training, millions to billions of parameters need to be communicated among workers at each iteration, and the expensive communication cost becomes a bottleneck.

To address the communication bottleneck, a wide variety of lossy gradient compression optimizers have been proposed to lower the communication volume. These algorithms can be typically divided into three groups, including low-precision approximation (*e.g.*, *1-bit SGD*(Seide et al. (2014)), *SignSGD*(Bernstein et al. (2018)), *TernGrad* (Wen et al. (2017)), and *QSGD*) (Alistarh et al. (2017)), *1-Bit Adam* (Tang et al. (2021))), low-rank simplification (*e.g.*, *ATOMO*(Wang et al. (2018)), *PowerSGD* (Vogels et al. (2019)), and *GradZip* (Cho et al. (2019))), and sparsification (*e.g.*, *Random-k* (Stich et al. (2018)), *Top-k* (Aji & Heafield (2017)), and *MSTop-k* (Shi et al. (2021))).

While much of the research on gradient compression algorithms has focused mainly on the high compression ratio, a more important yet underexplored problem is how to decrease the actual system-level runtime and increase the distributed scaling efficiency. Actually, some recent works (Xu et al. (2020),Agarwal et al. (2022)) pointed out, when distributedly training typical models (*e.g.*, ResNet-50 and BERT-Base) with off-the-shelf *DistributedDataParallel (DDP)* at typical bandwidths (*e.g.*, 10Gbps), these existing gradient compression algorithms with high compression ratios are

still **slower** than the original uncompressed optimizers. This is because they exhibit one or more of the following weaknesses (Xu et al. (2020),Agarwal et al. (2022)): ($i$) Some gradient compression algorithms should perform compression/decompression and communication within the limited time frame, and the time cost of compression/decompression, in some cases, is close to and even larger than the savings by the reduces communications; ($ii$) some gradient compression algorithms cannot take full advantage of overlapping between gradient computation and communication. Because if the gradient computation and compression/decompression overlap, their intensive computation will compete with each other for GPU resources, which can result in an overall slowdown; ($iii$) due to inherent structures, some algorithms can only use inefficient collective communication primitive, such as *All-Gather*; ($iv$) some gradient compression algorithms need to harness uncompressed optimizers to warm up at the early stage. The warm-up time is commonly nontrivial which to some extent renders their high compression ratios vacuous. Therefore, from a system-level perspective, the design ethos of a system-efficient communication-compression algorithm is that we should guarantee that the compression/decompression of the algorithm is computationally light and takes less time, and that the corresponding communication should also be friendly to efficient collective communication primitives. Additionally, there is no need to resort to an uncompressed optimizer for warm-up.

To this end, we propose a communication-compression optimization algorithm, referred to as Binary SGD-Momentum (*BinSGDM*), in which the core updating rule is $x_{t+1} = x_t - \alpha_t \mathcal{Q}\left(\frac{m_t}{b_t}\right)$ where $m_t = \beta m_{t-1} + (1-\beta)g_t$ , $b_t = \beta b_{t-1} + (1-\beta)|g_t|$ and $g_t$ is the gradient, and $\mathcal{Q}(\cdot)$ is a binary quantization operator. The main difference between *BinSGDM* and existing gradient-quantization algorithms is that we directly quantize the entire update $\frac{m_t}{b_t}$ rather than quantize the gradient $g_t$ or the momentum $m_t$. Due to $-1 \leq \frac{(m_t)_j}{(b_t)_j} \leq 1$ where $(m_t)_j, (b_t)_j$ are the $j^{th}$ element of $m_t, b_t$ , each element of $\frac{m_t}{b_t}$ is easy to be randomly quantized to $1$ or $-1$ in probability, so the quantization is computationally light. Another advantage of *BinSGDM* is that it does not need a full-precision optimizer to warm up at the early stage to ensure stable convergence. Besides, we theoretically demonstrate *BinSGDM*'s convergence rate can match that of the original *Adam*. Moreover, according to the nature of *BinSGDM*, we specifically devise an efficient hierarchical communication scheme to further speed up communication, which sufficiently leverages the ultra-high intra-bandwidth among GPUs within the same node and efficient commutation primitives rather than *All-Gather*.

In particular, we make the following key **contributions**:

- We propose a novel communication-compress distributed optimizer, dubbed *BinSGDM*. **To the best of our knowledge, it is the first algorithm that quantizes the entire model update of an adaptive optimizer and does not need to leverage uncompressed optimizers to warm up to address the convergence issue**, which makes compression/decompression computationally light and the extreme quantization ratio exert its best function (Section 2).

- We theoretically prove that even though extreme 1-bit quantization is employed, ***BinSGDM* still promise the same convergence speed as the full-precision *Adam*** (Section 3).

- We present a new hierarchical communication scheme for 1-bit communication, called *Hierarchical 1-bit All-Reduce*, **which sufficiently harnesses the ultra-fast intra-connects to accelerate the local communication, and utilize more efficient commutation primitives to further reduce the communication overhead** (Section 4).

- We perform extensive distributed training experiments to demonstrate the effectiveness of the proposed algorithm. **As far as we know, our algorithm is the first work to consistently trump the uncompressed optimizers with the highly system-level optimized *DDP* in overall running time at no inference performance cost**, reaching up to $2.47\times$ speedup for Resnet-50 and $6.26\times$ speedup for BERT-Base on $64$ GPUs. The better scalability makes *BinSGDM* promising to train more large-scale models (Section 5).

## 2  EXTREMELY ONE-BIT QUANTIZED BINSGDM

In this section, we focus on solving the following problem when training a DNN model distributedly:

$$\min_{x \in \mathbb{R}^d} f(x) = \frac{1}{n}\sum_{i=1}^{n} f_i(x; \xi^{(i)}) \tag{1}$$

where $x$ is the $d$-dimensional model parameter, $n$ is the number of distributed workers. $\xi^{(i)}$ is the sampled min-batch data on the $i$-the worker. The sampled min-batch data on all the workers is independent and identically distributed (*i.i.d.*). $f_i(x; \xi^{(i)})$ is the loss function. Note that $f_i(x; \xi_i)$ is commonly abbreviated as $f_i(x)$ in the following.

When we directly employ vallina full-precision and dense-computation optimizers to train a large-scale DNN model on distributed workers, the gradient communication among workers at each iteration becomes a bottleneck. Elegant *SignSGD* was proposed to alleviate the bottleneck problem, which merely takes the sign of each coordinate of the gradients. Although it can substantially reduce the communication overhead, its practical performance is still inferior to popular optimizers, such as *Adam*. Fortunately, we observe that the mathematical formulations of *SignSGD* and *Adam* have close connections, so it leaves us an opportunity to propose a new optimizer that can combine their merits, *i.e.*, considerably reducing the communication volume with light computation yet maintaining fast convergence speed and high inference performance.

The mathematical update step of *SignSGD* can be formulated as:

$$x_{t+1} \leftarrow x_t - \alpha_t \text{Sign}(g_t) = x_t - \alpha_t \frac{g_t}{|g_t|} \qquad (2)$$

where $\alpha_t$ is the learning rate, $g_t$ denotes the estimated unbias noisy gradient of $f(x_t)$ with random samples. Note that the divider here is an element-wise divider.

Whereas the updating rule of vallina *Adam* can be expressed as:

$$
\begin{aligned}
m_t &\leftarrow \beta_1 m_{t-1} + (1 - \beta_1) g_t, \\
v_t &\leftarrow \beta_2 v_{t-1} + (1 - \beta_2) g_t^2, \\
x_{t+1} &\leftarrow x_t - \alpha_t \frac{m_t}{\sqrt{v_t}},
\end{aligned}
\qquad (3)
$$

where $\beta_1$ and $\beta_2$ represents the exponential moving average factors [1].

If taking the exponential moving average factors to zero, $\beta_1, \beta_2 \rightarrow 0$, in Eq. (3), *Adam* will be equal to *SignSGD*.

Given the observations above, we propose a new optimizer that is an intermediate between *SignSGD* and *Adam*, referred to as *BinSGDM*, *i.e.*,

$$
\begin{aligned}
m_t &\leftarrow \beta m_{t-1} + (1 - \beta) g_t, \\
b_t &\leftarrow \beta b_{t-1} + (1 - \beta) |g_t|, \\
x_{t+1} &\leftarrow x_t - \alpha_t \mathcal{Q}\left(\frac{m_t}{b_t}\right),
\end{aligned}
\qquad (4)
$$

where the $j$-th elements of $m_t, b_t$ rigorously satisfies $-1 \leq \frac{(m_t)_j}{(b_t)_j} \leq 1$, $\mathcal{Q}(\cdot)$ is an element-wise quantization operator, and it quantizes the $j$-th element of $\frac{m_t}{b_t}$ as follows:

$$
\mathcal{Q}\left(\frac{(m_t)_j}{(b_t)_j}\right) = \begin{cases} 1, & \text{with probability } p = \frac{1}{2}\left(\frac{(m_t)_j}{(b_t)_j} + 1\right) \\ -1, & \text{with probability } 1 - p \end{cases}, \qquad (5)
$$

where $\mathbb{E}\left(\mathcal{Q}\left(\frac{m_t}{b_t}\right)\right) = \frac{m_t}{b_t}$, so $\mathcal{Q}(\cdot)$ is unbiased.

The detailed implementation of *BinSGDM* in a parameter-server model is illustrated in Algorithm 1. Some appealing characters of *BinSGDM* are summarized in the following:

- All the existing communication-efficiency optimizers are built upon gradient compression. In contrast, to the best of our knowledge, we are the first to directly quantize the entire model update, which will streamline the quantization. Moreover, each element of $\frac{m_t}{b_t}$ bounds

---

[1]For simplicity, we omit the bias correction for $m_t$ and $v_t$ and the small constant in the numerator.

---

**Algorithm 1.** BinSGDM

1: **Input**: all workers's model parameter $x_0, x_1$, the $i^{th}$ worker's momentum $m_0^{(i)} = 0$, $b_0^{(i)} = 0$, the $i^{th}$ worker's local error $e_0^{(i)} = 0$, server's global error $\bar{e}_0 = 0$, exponential moving average factor $\beta$, the threshold $T_0$, and the learning rate sequence $\{\alpha_t\}$.

2: **for** $t = 1, ..., T$ **do**

3:     (**On the $i^{th}$ worker**)

4:     Randomly sample $\xi_t^{(i)}$ and compute local gradient: $g_t^{(i)} = \nabla f_i(x_t; \xi_t^{(i)})$

5:     Update the local $m_t^{(i)}$: $m_t^{(i)} = \beta m_{t-1}^{(i)} + (1 - \beta)g_t^{(i)}$

6:     Update the local $\hat{b}_t^{(i)}$: $\hat{b}_t^{(i)} = \beta \hat{b}_{t-1}^{(i)} + (1 - \beta)|g_t^{(i)}|$

7:     Update the local $b_t^{(i)}$: **if** $t > T_0$ $\{ b_t^{(i)} = \max(b_{t-1}^{(i)}, \hat{b}_t^{(i)}) \}$ **else** $\{ b_t^{(i)} = \hat{b}_t^{(i)} \}$ *

8:     Quantize the local update: $u_t^{(i)} = \mathcal{Q}(\frac{m_t^{(i)}}{b_t^{(i)}} + e_{t-1}^{(i)})$

9:     Update the local error feedback $e_t^{(i)}$: $e_t^{(i)} = e_{t-1}^{(i)} + \frac{m_t^{(i)}}{b_t^{(i)}} - u_t^{(i)}$

10:    Send $u_t^{(i)}$ to the server

11:    (**On server**)

12:    Average all received $q_t$ and quantize it: $\bar{u}_t = \mathcal{Q}(\frac{1}{n}\sum_{i=1}^{n} u_t^{(i)} + \bar{e}_{t-1})$

13:    Update the global error feedback $\bar{e}_t$: $\bar{e}_t = \bar{e}_{t-1} + \frac{1}{n}\sum_{i=1}^{n} u_t^{(i)} - \bar{u}_t$

14:    Send back $\bar{u}_t$ to all workers

15:    (**On the $i^{th}$ worker**)

16:    Update the local model parameter $x_{t+1}$: $x_{t+1} = x_t - \alpha_t \bar{u}_t$

17: **end for**

---

\* This step follows the technique in AMSGrad Reddi et al. (2018). It is more about theoretical significance, and we commonly do not implement it in practice.

in the range $[-1, 1]$, and then it is extremely quantized to binary $1$ or $-1$. Thus, the quantization in *BinSGDM* is computed easily and lightly, compared to the existing gradient-quantized optimizers [2] .

- Unlike *SignSGD*, *Adam* adaptively preconditions the gradients with $v_t$, which is the key to ensuring a fast convergence rate in practice. *BinSGDM* also inherits the character of adaptive preconditioning from *Adam* to accelerate the convergence speed. The difference is that we keep the exponential moving average factor for $m_t$ and $b_t$ the same, so that each element of $\frac{m_t}{b_t}$ bounding in the range $[-1, 1]$ can be strictly guaranteed, which is crucial to perform light quantization.

- When the model parameter $x_t$ is close to a local optimal value during training, an ideal optimizer should make sure the updates gradually decay to zero, otherwise, $x_t$ will oscillate around the optimum and cannot indeed approach it. *SignSGD* and *Adam* do not have this appealing property, while, as for unquantized *BinSGDM* (we refer to it as *SoftSignSGD*, and the implementation details for it please see Algorithm 2 in the Appendix), the gradient $g_t$ will continually change its sign around the optimum of which the gradient is zero, so that the update $\frac{m_t}{b_t}$ will damp to zero. An Example illustrating this phenomenon is provided in Section C in the Appendix.

**Remark.** We have noticed that the prior works *1-bit Adam* (Tang et al. (2021)) and its variants (Li et al. (2021), Lu et al. (2022)) also quantize the communication data to 1-bit. However, the design ethos of *1-bit Adam* and the proposed *BinSGDM* are completely different. *1-bit Adam* is still built on gradient compression rather than the entire update. The motivation of *1-bit Adam* is to harness the error feedback (EF) technique (Seide et al. (2014), Stich et al. (2018)) to compensate for the gradient information lost to alleviate the convergence issue. However, unlike *SGD*, the parameter update in *Adam* no longer linearly depends on the gradient, so that EF cannot be directly employed. Authors of *1-bit Adam* observed that *Adam*'s variance (non-linear term) becomes stable after the early stage, *1-bit Adam* runs full-precision *Adam* in the beginning (warmup phase) and utilizes it

---

[2]The typical gradient-quantized optimizer *QSGD* quantizes the gradient as follows:

$$\mathcal{Q}((g_t)_j) = \begin{cases} \|g_t\|_2 \text{sign}((g_t)_j) \cdot \frac{r}{s}, & \text{with probability } p_i = \frac{s|(g_t)_j|}{\|g_t\|_2} - r \\ \|g_t\|_2 \text{sign}((g_t)_j) \cdot \frac{r+1}{s}, & \text{with probability } 1 - p_i \end{cases}$$

where $0 \leq r < s$ $(r, l \in \mathbb{N})$ and $\frac{s|(g_t)_j|}{\|g_t\|_2} \in [\frac{r}{s}, \frac{r+1}{s}]$.

as a precondition for *SGDM* during the rest of training (compression phase), and then EF in the compression phase help *1-bit Adam* to converge as rapidly as uncompressed *Adam*. There are two aspects that influence *1-bit Adam* to indeed accelerate communication. *First*, the warmup steps commonly make up 15%-25% of the total steps in *1-bit Adam*, which to some extent discounts the high quantization ratio. *Second*, in the compression phase, *1-bit Adam* should communicate the signs of the gradients of each layer as well as the average scale of all the gradients in this layer, while in *DDP*, raising communication efficiency, the gradients from many layers should be flatted to 1-dimension and concatenated together, and then packed to one bucket to communicate, which means the scale factor of each layer cannot be computed. Therefore, *1-bit Adam* is not compatible with *DDP*, which will also lower communication efficiency.

## 3 THEORETICAL ANALYSIS

In this section, we present the theoretical convergence guarantee for *BinSGDM* (Algorithm 1). We first introduce some necessary assumptions.

**Assumption 1.**[Bounded infimum] *For any $x$ and a constant $f^*$, we have the objective value $f(x) \geq f^*$.*

**Assumption 2.** [Lipschitz continuous gradient] *The gradient $\nabla f(\cdot)$ is L-Lipschitz continuous, i.e., , $\|\nabla f(x) - \nabla f(y)\| \leq L\|x - y\|_2, \quad \forall x, y \in \mathbb{R}^d$.*

**Assumption 3.** [Unbias and indpendent noisy gradient] *The gradient with respect to the random samples on each worker and at a different time is independent identically distributed (i.i.d.), i.e., $\mathbb{E}[g_t^{(i)}] = \nabla f(x_t), \forall t \geq 1$, $g_t^{(i)}$ is independent of $g_t^{(j)}$ for $i \neq j$, and $g_{t_1}^{(i)}$ is independent of $g_{t_2}^{(j)}$ for $t_1 \neq t_2$.*

**Assumption 4.** [Bounded gradient] *The noisy gradient and the full-set gradient are bounded i.e., $\|g_t^{(i)}\| \leq G, \quad \|\nabla f_t(x)\| \leq G, \quad \forall t \geq 1$.*

Under the assumptions above, we then present the theoretical convergence for BinSGDM in Algorithm 1.

**Theorem 1.** *For BinSGDM in Algorithm 1, under Assumption 1-4, assuming $(b_t^{(i)})_j \geq \rho > 0$, $\forall j \in [1, 2, ..., d]^3$, choosing $\alpha_t = \frac{c}{\sqrt{t}}$, $\forall t \in [1, 2, ..., T]$ and $\alpha_0 = \alpha_1$, and defining $z_1 = x_1 + \alpha_1(\delta_1 - e_1)$ where $\delta_1 = \frac{1}{n}\sum_{i=1}^n \frac{m_1^{(i)}}{b_1^{(i)}} - \frac{\sum_{i=1}^n m_1^{(i)}}{\sum_{i=1}^n b_1^{(i)}}$ and $e_1 = \frac{1}{n}\sum_{i=1}^n e_1^{(i)} + \bar{e}_1$, we then have the following*

$$\mathbb{E}\left[\frac{1}{T}\sum_{t=1}^T \|\nabla f(x_t)\|\right]^2 \leq \frac{C_1}{\sqrt{T}} + \frac{C_2(1 + \log T)}{\sqrt{T}}, \tag{6}$$

*where*

$$C_1 = cG\left(\mathbb{E}[f(z_1) - f^*] + \frac{3c^2 dL}{16} + \frac{\beta cdG^2}{(1-\beta)\rho} + \frac{4cdG^2}{\rho} + \frac{c^2\beta^2 LG^2 d}{\rho^2(1-\beta)^2}\right),$$

$$C_2 = c^3 G\left(\frac{(8\beta^2 + 10\beta + 5)L^2 d}{(1-\beta)^2} + \frac{G^2(1+L)}{2\rho^2} + 2dL\right).$$

## 4 HIERARCHICAL 1-BIT ALL-REDUCE

The data in *BinSGDM* to communicate is one-bit, so it cannot be directly aggregated through the efficient *All-Reduce*. Moreover, the intra-node bandwidth and inter-node bandwidth are severely imbalanced. If we aggregate the data uniformly from intra-nodes and inter-nodes, the communication will be slowed down by the inter-node data exchanges.

In light of the problems above, we propose a hierarchical communication scheme, called *Hierarchical 1-bit All-Reduce*, to efficiently aggregate our 1-bit data, which can hierarchically take advantage

---

3We commonly add a small constant to $b_t$ to avoid zero denominators for numerical stability, which guarantees this assumption holds in practice.

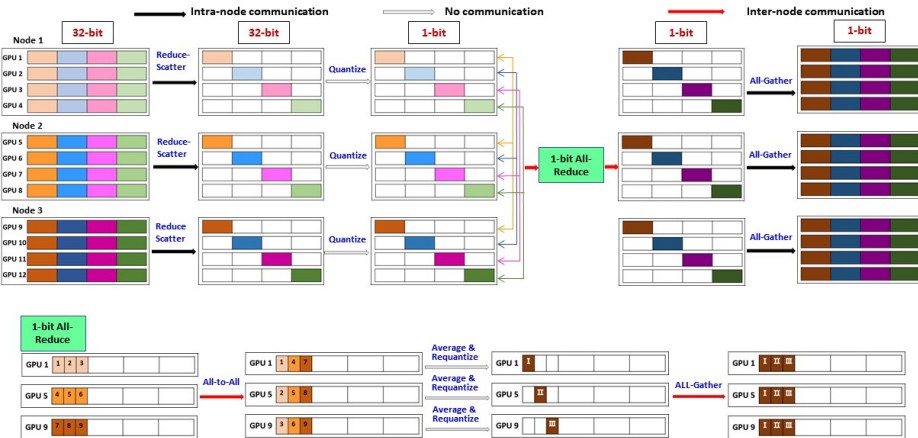

Figure 1: Paradigm of Hierarchical 1-bit All-Reduce

of the ultra-high intra-node bandwidth and reduce the inter-node communication overhead. Assuming we have $n$ nodes and each node contains $m$ GPUs and the overall volume for each GPU needs to communicate is $P$, as visually illustrated in Figure 8, the steps of *Hierarchical 1-bit All-Reduce* is as follows: ($i$) Each GPU conducts *Reduce-Scatter* to locally aggregate the scatter the data within a node, and the communication volume for each GPU is $\frac{(m-1)P}{m}$. ($ii$) Each GPU performs *BinSGDM* to quantize the data, and then volume becomes $\frac{P}{32m}$ on each GPU. ($iii$) Each GPUs conducts *1-bit All-Reduce* to inter-aggregate data. This step includes two sub-steps: 1) Each GPU performs *All-to-All* to collect the data of corresponding GPUs in other nodes, and the communication volume is $\frac{(n-1)P}{32mn}$; 2) each GPU averages and re-quantizes the data, and then conducts *All-Gather* to gather the data, and the communication volume is also $\frac{(n-1)P}{32mn}$. (iv) Each GPU performs *All-Gather* to intra-aggregate data, and the communication volume is $\frac{(m-1)P}{32m}$.

Compared to the time cost of inter-node communication, the time cost of inter-node communication is trivial. Hence, when leveraging *Hierarchical 1-bit All-Reduce*, the most communication cost comes from *1-bit All-Reduce* in Step ($iii$), and then the communication volume across nodes for all GPUs is approximately $\frac{2(n-1)P}{32}$. In contrast, if we simply utilize the original *All-Gather* to aggregate data, the communication volume across nodes for all GPUs is approximately $\frac{m^2 n(n-1)P}{32}$. Therefore, *Hierarchical 1-bit All-Reduce* is considerably more efficient than the original *All-Gather*.

## 5 EXPERIMENTS

Table 1: System throughput and Test Accuracy of *SGDM*, *1-bit Adam* and *BinSGDM* for training ResNet-50 on ILSVRC2012 from scratch with $8, 16, 32, 64$ GPUs.

| Optimizer | #GPUs | 32 samples per GPU | | 128 samples per GPU | |
|---|---|---|---|---|---|
| | | Throughput (samples / s) | Top-1 Acc. (%) | Throughput (samples / s) | Top-1 Acc.(%) |
| SGDM | 8 | 3693 (1.00×) | 76.19 | 5272 (1.00×) | 75.05 |
| 1-bit Adam | | 3243 (0.83×) | 75.55 | 5229 (0.99×) | 75.42 |
| BinSGDM | | 3462 (0.94×) | 75.98 | 5251 (0.99×) | 75.45 |
| SGDM | 16 | 2959 (1.00×) | 75.96 | 6189 (1.00×) | 74.61 |
| 1-bit Adam | | 4745 (1.60×) | 75.33 | 8836 (1.42×) | 75.05 |
| BinSGDM | | 6015 (2.03×) | 75.53 | 9633 (1.56×) | 75.09 |
| SGDM | 32 | 4270 (1.00×) | 75.47 | 9909 (1.00×) | 74.54 |
| 1-bit Adam | | 7268 (1.70×) | 75.18 | 13827 (1.40×) | 74.82 |
| BinSGDM | | 9416 (2.21×) | 75.27 | 15950 (**1.61×**) | 74.82 |
| SGDM | 64 | 6189 (1.00×) | 75.37 | 16640 (1.00×) | 74.22 |
| 1-bit Adam | | 5546 (0.89×) | 75.54 | 16426 (0.99×) | 74.34 |
| BinSGDM | | 15253 (**2.47×**) | 75.30 | 23727 (1.43×) | 74.24 |

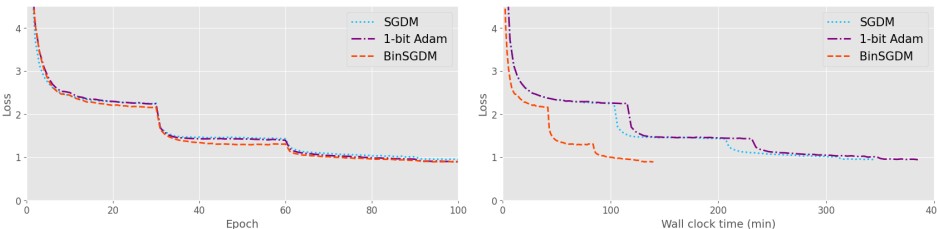

(a) Epoch-wise, ResNet-50, batch size=$32 \times 64$    (b) Time-wise, ResNet-50, batch size=$32 \times 64$

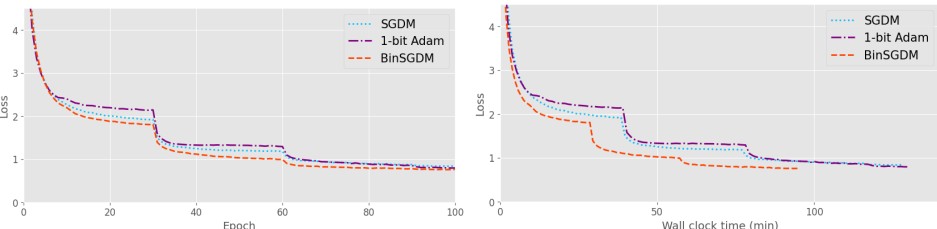

(c) Epoch-wise, ResNet-50, batch size=$128 \times 64$(d) Time-wise, ResNet-50, batch size=$128 \times 64$

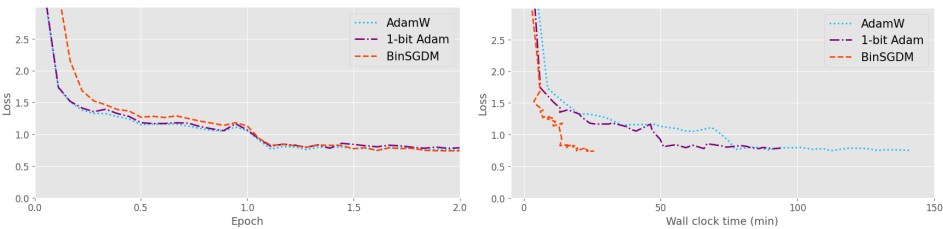

(e) Epoch-wise, Bert-Base, batch size=$3 \times 64$    (f) Time-wise, Bert-Base, batch size=$3 \times 64$

Figure 2: Epoch-wise and time-wise convergence speed for training ResNet-50 with 32 samples per GPU, ResNet-50 with 128 samples per GPU, and fine tuning BERT-Base with 3 samples per GPU with 64 GPUs.

Table 2: System throughput and F1-Score / Excat-Match of *AdamW*, *1-bit Adam* and *BinSGDM* for fine tuning BERT-base on SQuAD 1.1 with 8, 16, 32, 64 GPUs.

| Optimizer | #GPUs | Throughput (samples/s) | F1-Score (%) | Exact Match (%) |
|---|---|---|---|---|
| AdamW | | 413 (1.00×) | 88.13 | 80.59 |
| 1-bit Adam | 8 | 358 (0.87×) | 88.05 | 80.06 |
| BinSGDM | | 412 (1.00×) | 88.71 | 81.18 |
| AdamW | | 84 (1.00×) | 88.47 | 81.07 |
| 1-bit Adam | 16 | 213 (2.54×) | 87.87 | 80.31 |
| BinSGDM | | 431 (5.13×) | 88.31 | 80.80 |
| AdamW | | 119 (1.00×) | 88.38 | 80.94 |
| 1-bit Adam | 32 | 274 (2.30×) | 87.78 | 80.08 |
| BinSGDM | | 730 (6.13×) | 88.08 | 80.50 |
| AdamW | | 158 (1.00×) | 88.13 | 80.94 |
| 1-bit Adam | 64 | 252 (1.59×) | 87.33 | 79.67 |
| BinSGDM | | 990 (**6.26×**) | 88.28 | 80.75 |

Recently, some recent works [Xu et al. (2020),Agarwal et al. (2022)] have demonstrated that when running with the system-level optimized distributed data-parallel frameworks(*e.g.*, *DDP* ), the existing typical communication-compression optimizers (not including *1-bit Adam*) runs still slower than the full-precision original *SGD/Adam* (the reasons please refer to the section of Introduction). Hence, we only evaluate the performances of *BinSGDM*, the uncompressed original *SGDM /AdamW*, and the closely relevant *1-bit Adam* based on performing distributed training experiments with the benchmark model ResNet-50 (CNN) and BERT-Base (Transformer). We show that running with the distributed data-parallel framework *DDP* in Pytorch, *BinSGDM* with the proposed specific commutation scheme is up to $2.47$ times faster for ResNet-50 and $6.26$ times faster for BERT-Base than the uncompressed optimizers with highly system-level optimized *all-reduce* at no accuracy cost.

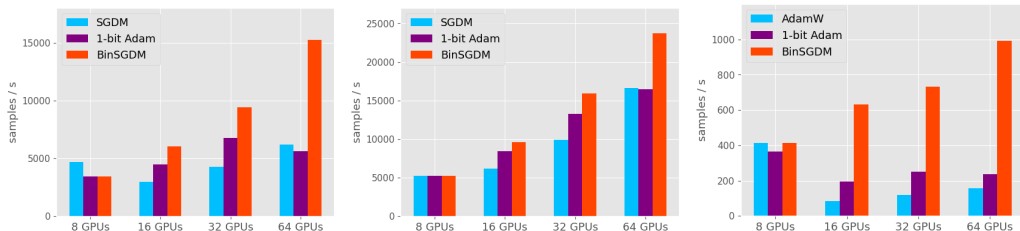

(a) ResNet-50, 32 samples / GPU  (b) ResNet-50, 128 samples / GPU  (c) BERT-Base, 3 samples / GPU

Figure 3: System throughput of optimizers for training (a) ResNet-50 with 32 samples per GPU, (b)ResNet-50 with 128 samples per GPU, and (c) fine tuning BERT-Base with 3 samples per GPU with 8, 16, 32, 64 GPUs.

## 5.1 EXPERIMENTAL SETTINGS

**Testbed.** Our experiments are implemented on {1, 2, 4, 8} nodes connected with 10Gbps Ethernet, and each node is equipped with 8 Nvidia Tesla A100-80GB GPUs. The hardware and software are the same in all instances. The operating system in each node is Ubuntu 20.04.4 LTS. Our experiments are performed in Pytorch 1.11.0, and other related libraries are CUDA-11.6, cuDNN-8.2, NCCL-2.10.3 and Pytorch 1.11.0. Notably, to be compatible with *DDP* of Pytorch, parts of *BinSGDM* and our hierarchical communication scheme are implemented in the customized communication hook of *DDP* in Pytorch.

**Training details.** For the experiments over ResNet-50, we evaluate the convergence and performance of *SGDM*, *1-bit Adam* and *BinSGDM* on ILSVRC2012. The batch size per GPU is set to 32 or 128 with the standard input resolution $224 \times 224$. When employing *SGDM (baseline)*, the learning rate starts at $0.1 \times \frac{batch\ size}{256}$ with momentum of 0.9 and weight decay of 0.0001. When employing *1-bit Adam* and *BinSGDM*, the learning rate starts at $0.001 \times \frac{batch\ size}{256}$ with weight decay of 0.0001, and $[\beta_1, \beta_2]$ for *1-bit Adam* is set to $[0.9, 0.999]$ and $\beta$ for *BinSGDM* is set to 0.95. Then, the learning rate is divided by 10 after 30, 60 and 90 epochs, and training is finally terminated after 100 epochs. Specifically, the first 15 epochs are used as the warmup stage for *1-bit Adam*. For the experiments over BERT-Base, we access the convergence and performance of *AdamW* (baseline), *1-bit Adam* and *BinSGDM* for SQuAD 1.1 fine-tuning task using a pre-trained BERT-Base model checkpoint from HuggingFace [4]. The batch size per GPU is set to 3. We perform fine-tuning for 2 epochs. The learning rate linearly increases to $0.3\times$ steps in the early 500 steps and then linearly decreases to 0 in the rest iteration. Specifically, the first $0.2\times$ steps are used as the warmup stage for *1-bit Adam*. $[\beta_1, \beta_2]$ for *AdamW*, and *1-bit Adam* is set to $[0.9, 0.999]$ and $\beta$ for *BinSGDM* is set to 0.9.

## 5.2 EXPERIMENTAL RESULTS

Figure 2 shows the sample-wise and time-wise training convergence behaviors for *SGDM / AdamW* (baseline), *1-bit Adam* and *BinSGDM* with ResNet-50 and BERT-Base running on 64 GPUs. The experimental results demonstrate that *BinSGD* can achieve a similar sample-wise convergence rate to the baseline, while its actual speed is substantially faster than the baseline and *1-bit Adam*, up to about $2.5\times$ speedup for ResNet-50 with batch size 32 per GPU on and about $6.3\times$ speedup for BERT-Base fine-tuning on SQuAD 1.1.

Figure 3 presents the system throughput for different optimizers running with ResNet-50 and BERT-Base on 8 GPUs to 64 GPUs. When training on 8 GPUs, since computation rather than communication dominates the running time, the throughput performance for *BinSGDM* is slightly inferior to *SGDM* and *AdamW*, but when the number of GPUs is growing, *BinSGDM* consistently outperforms the counterparts, and the more the number of GPUs is, the more the superiority becomes obvious. Furthermore, the system throughput for *SGDM*, *AdamW* and *1-bit Adam* even decreases with GPUs increasing in some cases, whereas the throughput for *BinSGDM* steadily grows, which indicates that *BinSGDM* can offer better scalability efficiency.

In terms of inference performance for ResNet-50, we evaluate the Top-1 accuracy after training on ILSVRC2012 from scratch. As the inference performance for BERT-Base, we measure the F1-score

---

[4]https://github.com/huggingface/transformers

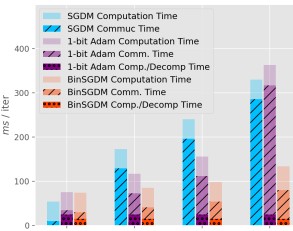 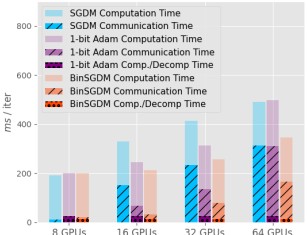 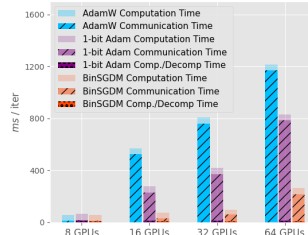

(a) ResNet-50, 32 samples / GPU    (b) ResNet-50, 128 samples / GPU    (c) BERT-Base, 3 samples / GPU

Figure 4: Computation time, communication time and compression/decompression time per iteration of optimizers for training (a) ResNet-50 with 32 samples per GPU, (b)ResNet-50 with 128 samples per GPU, and (c) fine tuning BERT-Base with 3 samples per GPU with 8, 16, 32, 64 GPUs.

and exact-match score after fine-tuning on SQuAD 1.1. As shown in Table 1, when the batch size is set to 32 samples per GPU, the accuracy for *BinSGDM* is slightly lower than *SGDM*. For CNN architecture, some works (Keskar & Socher (2017), Zhou et al. (2020)) pointed out that adaptive optimizers commonly generalize worse than *SGDM*. However, when the batch size becomes larger (Table2), *1-bit Adam* and *BinSGDM* achieve better accuracy. The reason for it is that a certain level of noise can be helpful for generalization (Smith & Le (2018)), biasing the optimizer towards wider valleys. Large batch size will reduce the randomness, while quantization errors in *BinSGDM* increase the randomness. Table 2 exhibits *BinSGDM* obtains similar or higher F1-score and exact-match score, compared to *AdamW* and *1-bit Adam*, which validates the inferencing effectiveness of the proposed *BinSGDM*.

## 5.3 COMMUNICATION EFFICIENCY ANALYSIS

As illustrated in Figure 4, when we perform training on a single node, the baseline full-precision *SGDM* and *Adam* is slightly faster than *BinSGDM* and *1-bit Adam*. Since the inter-GPU bandwidth within a node is ultra-high, the communication time becomes negligible, and the newly introduced compression/decompression by communication-compression optimizers takes up extra time. Due to the light-computation quantization, *BinSGDM* with ResNet-50 and BERT-Base only takes about $15ms$ and $8ms$ respectively for compression/decompression. When we perform distributed training on two nodes, the bandwidth between nodes is limited (10Gbps in our experimental testbed), and the communication time should be reckoned with. For the uncompressed *SGDM* and *Adam*, the communication time considerably exceeds the computation time for ResNet with 32 samples per GPU and BERT-Base, which makes the system throughput even lower than that on a single node (shown in Figure 2). Since extreme quantization in *BinSGDM* substantially reduces the communication overhead (up to $32\times$ reduction), the compunction time for *BinSGDM* increases slightly. When the number of nodes is further increasing, a good communication scheme becomes significant. Leveraging our proposed *Hierarchical 1-bit All-Reduce*, the overall inter-node communication volume exchanged is proportional to the number of nodes, while, for *CompressedAllreduce* utilized by *1-bit Adam*, the overall communication volume exchanged among nodes is proportional to the number of GPUs (8 times number of nodes in our experiment). Therefore, with the number of nodes increasing, the communication time for *BinSGDM* raises gently, but the communication time for *1-bit Adam* grows abruptly.

## 6 CONCLUSION

In this work, we present a novel communication compression optimizer for distributed training. The optimizer is not only easy and light to compute but also quantizes the communication data to an extreme one bit. We also theoretically demonstrate that *BinSGDM* can converge as fast as the original Adam. To further accelerate the communication. will specifically present a novel communication scheme for *BinSGDM* to replace the inefficient naive *All-Gather*. Extensive experiments on training the benchmark ResNet-50 and BERT-Base have validated the effectiveness and efficiency of *BinSGDM* over the uncompressed *SGD,Adam* and the most relevant *1-bit Adam*.

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

## A THEORETICAL ANALYSIS FOR ALGORITHM 1

In practice, we implement *BinSGDM* in a non-parameter-server model to further reduce the communication overhead, but the data exchange is essentially equivalent to that in a parameter-server prototype. Hence, we provide the theoretical analysis for *BinSGDM* in a parameter-server model as shown in Algorithm 1.

According to Algorithm 1, the update $\bar{u}_t$ can be recursively formulated as

$$
\begin{aligned}
\bar{u}_t =& \mathcal{Q}\left(\frac{1}{n}\sum_{i=1}^{n} u_t^{(i)} + \bar{e}_t\right) \\
=& \frac{1}{n}\sum_{i=1}^{n} u_t^{(i)} + \bar{e}_t - \bar{e}_{t+1} \\
=& \frac{1}{n}\sum_{i=1}^{n} \mathcal{Q}\left(\frac{m_t^{(i)}}{b_t^{(i)}} + e_t^{(i)}\right) + \bar{e}_t - \bar{e}_{t+1} \\
=& \frac{1}{n}\sum_{i=1}^{n}\left(\frac{m_t^{(i)}}{b_t^{(i)}} + e_t^{(i)} - e_{t+1}^{(i)}\right) + \bar{e}_t - \bar{e}_{t+1} \\
=& \frac{1}{n}\sum_{i=1}^{n}\frac{m_t^{(i)}}{b_t^{(i)}} + \frac{1}{n}\sum_{i=1}^{n}\left(e_t^{(i)} - e_{t+1}^{(i)}\right) + \bar{e}_t - \bar{e}_{t+1}
\end{aligned}
\tag{7}
$$

Denote

$$g_t \triangleq \frac{1}{n} \sum_{i=1}^{n} g_t^{(i)}, \tag{8}$$

$$m_t \triangleq \frac{1}{n} \sum_{i=1}^{n} m_t^{(i)} = \beta m_{t-1} + (1 - \beta) g_t, \tag{9}$$

$$b_t \triangleq \frac{1}{n} \sum_{i=1}^{n} b_t^{(i)}, \tag{10}$$

$$\delta_t \triangleq \frac{1}{n} \sum_{i=1}^{n} \frac{m_t^{(i)}}{b_t^{(i)}} - \frac{m_t}{b_t}, \tag{11}$$

$$e_t \triangleq \frac{1}{n} \sum_{i=1}^{n} e_t^{(i)} + \bar{e}_t \tag{12}$$

$$\tag{13}$$

Hence, the updating rule can be summarized as

$$x_{t+1} = x_t - \alpha_t \bar{u}_t$$
$$= x_t - \alpha_t \left( \frac{m_t}{b_t} + \delta_t + e_t - e_{t+1} \right) \tag{14}$$

## A.1 AUXILIARY LEMMAS

**Lemma 1.** *Let $u_t = \frac{m_t}{b_t}$, the element-wise quantization function is defined in Eq.(5) can be reformulated as*

$$\mathcal{Q}\left((u_t)_j\right) = \begin{cases} 1, & \text{with probability } p = \frac{(u_t)_j + 1}{2} \\ -1, & \text{with probability } 1 - p \end{cases} \quad (j \in \{1, 2, ..., d\}, \ -1 \le (u_t)_j \le 1). \tag{15}$$

*We have $e_t = u_t - \mathcal{Q}(u_t)$, and then the following holds true*

$$\mathbb{E}[e_t] = 0, \ \mathbb{E}\left[\|e_t\|^2\right] \le d. \tag{16}$$

**Proof.** From Eq.(15), we know

$$\mathbb{E}[(e_t)_j] = \mathbb{E}[u_t - \mathcal{Q}(u_t)]$$
$$= \frac{1}{2}((u_t)_j + 1)((u_t)_j - 1) + (1 - \frac{1}{2}((u_t)_j + 1))((u_t)_j + 1) = 0, \tag{17}$$

and,

$$\mathbb{E}\left[(e_t)_j^2\right] = \mathbb{E}\left[((u_t)_j - \mathcal{Q}((u_t)_j))^2\right]$$
$$= \frac{1}{2}((u_t)_j + 1)((u_t)_j - 1)^2 + (1 - \frac{1}{2}((u_t)_j + 1))((u_t)_j + 1)^2 \tag{18}$$
$$= 1 - ((u_t)_j)^2 \le 1.$$

Hence,

$$\mathbb{E}[e_t] = 0, \ \mathbb{E}\left[\|e_t\|^2\right] \le d. \tag{19}$$

**Lemma 2.** *Let $x_0 = x_1$ and $\alpha_0 = \alpha_1$ in Algorithm 1, defining the sequence*

$$z_1 = x_1 + \alpha_1(\delta_1 - e_1) \tag{20}$$

$$z_t = x_t + \frac{\beta}{1 - \beta}(x_t - x_{t-1}) + \frac{\alpha_{t-1}}{1 - \beta}(\delta_{t-1} + \beta e_{t-1} - e_t), \forall t \ge 2. \tag{21}$$

*Then the following equality will hold, i.e.,*

$$z_{t+1} = z_t + \frac{\beta}{1-\beta}\left(\frac{\alpha_{t-1}}{b_{t-1}} - \frac{\alpha_t}{b_t}\right) \odot m_{t-1} - \alpha_t \frac{g_t}{b_t} - \alpha_{t-1}\delta_{t-1} - (\alpha_t - \alpha_{t-1})e_t. \tag{22}$$

**Proof.** For $t = 1$, we have

$$
\begin{aligned}
z_2 - z_1 &= x_2 + \frac{\beta}{1-\beta}(x_2 - x_1) + \frac{\alpha_1}{1-\beta}(\delta_1 + \beta e_1 - e_2) - (x_1 + \alpha_1(\delta_1 - e_1)) \\
&= (\frac{\beta}{1-\beta} + 1)(x_2 - x_1) + \frac{\alpha_1}{1-\beta}(\delta_1 + \beta e_1 - e_2) - \alpha_1(\delta_1 - e_1) \\
&= -\frac{\alpha_1}{1-\beta}\left(\frac{(1-\beta)g_1}{b_1} + \delta_1 + e_1 - e_2\right) + \frac{\alpha_1}{1-\beta}(\delta_1 + \beta e_1 - e_2) - \alpha_1(\delta_1 - e_1) \\
&= -\alpha_1 \frac{g_1}{b_1} - \alpha_0 \delta_1
\end{aligned}
\tag{23}
$$

where the second equality follows the updating rule in Eq.(14).

For $t \geq 2$, following the updating rule in Eq.(14), we have

$$
\begin{aligned}
x_{t+1} - x_t + \alpha_t(\delta_t + e_t - e_{t+1}) &= -\alpha_t \frac{m_t}{b_t} \\
&= -\alpha_t \frac{\beta m_{t-1} + (1-\beta)g_t}{b_t} \\
&= \beta\left(x_t - x_{t-1} + \alpha_{t-1}(\delta_t + e_{t-1} - e_t)\right) \\
&\quad + \beta\left(\frac{\alpha_{t-1}}{b_{t-1}} - \frac{\alpha_t}{b_t}\right) \odot m_{t-1} - (1-\beta)\alpha_t \frac{g_t}{b_t}
\end{aligned}
\tag{24}
$$

We know $x_{t+1} - x_t + \alpha_t(e_t - e_{t+1}) = (1-\beta)(x_{t+1} + -\alpha_t(e_{t+1} - \delta_t)) - (1-\beta)(x_t - \alpha_t e_t) + \beta(x_{t+1} - x_t + \alpha_t(\delta_t + e_t - e_{t+1}))$, so Eq. (24) can be rearranged as

$$
\begin{aligned}
&(1-\beta)(x_{t+1} + \alpha_t(\delta_t - e_{t+1})) + \beta(x_{t+1} - x_t + \alpha_t(\delta_t + e_t - e_{t+1})) \\
=&(1-\beta)(x_t - \alpha_t e_t) + \beta\left(x_t - x_{t-1} + \alpha_{t-1}(\delta_{t-1} + e_{t-1} - e_t)\right) \\
&+ \beta\left(\frac{\alpha_{t-1}}{b_{t-1}} - \frac{\alpha_t}{b_t}\right) \odot m_{t-1} - (1-\beta)\alpha_t \frac{g_t}{b_t}
\end{aligned}
\tag{25}
$$

Divided both sides by $1 - \beta$, we obtain

$$
\begin{aligned}
&x_{t+1} + \alpha_t(\delta_t - e_{t+1}) + \frac{\beta}{1-\beta}(x_{t+1} - x_t + \alpha_t(\delta_t + e_t - e_{t+1})) \\
=&x_t + \alpha_{t-1}(\delta_{t-1} - e_t) + \frac{\beta}{1-\beta}\left(x_t - x_{t-1} + \alpha_{t-1}(\delta_{t-1} + e_{t-1} - e_t)\right) \\
&+ \frac{\beta}{1-\beta}\left(\frac{\alpha_{t-1}}{b_{t-1}} - \frac{\alpha_t}{b_t}\right) \odot m_{t-1} \\
&- \alpha_t \frac{g_t}{b_t} - \alpha_{t-1}\delta_{t-1} - (\alpha_t - \alpha_{t-1})e_t
\end{aligned}
\tag{26}
$$

Rearranging Eq. (26), we have

$$
\begin{aligned}
&x_{t+1} + \frac{\beta}{1-\beta}(x_{t+1} - x_t) + \frac{\alpha_t}{1-\beta}(\delta_t + \beta e_t - e_{t+1}) \\
=&x_t + \frac{\beta}{1-\beta}(x_t - x_{t-1}) + \frac{\alpha_{t-1}}{1-\beta}(\delta_{t-1} + \beta e_{t-1} - e_t) \\
&+ \frac{\beta}{1-\beta}\left(\frac{\alpha_{t-1}}{b_{t-1}} - \frac{\alpha_t}{b_t}\right) \odot m_{t-1} \\
&- \alpha_t \frac{g_t}{b_t} - \alpha_{t-1}\delta_{t-1} - (\alpha_t - \alpha_{t-1})e_t
\end{aligned}
\tag{27}
$$

Define the sequence

$$z_t = x_t + \frac{\beta}{1-\beta}(x_t - x_{t-1}) + \frac{\alpha_{t-1}}{1-\beta}(\delta_{t-1} + \beta e_{t-1} - e_t) \tag{28}$$

We finally obtain

$$z_{t+1} = z_t + \frac{\beta}{1-\beta}\left(\frac{\alpha_{t-1}}{b_{t-1}} - \frac{\alpha_t}{b_t}\right) \odot m_{t-1} - \alpha_t \frac{g_t}{b_t} - \alpha_{t-1}\delta_{t-1} - (\alpha_t - \alpha_{t-1})e_t. \tag{29}$$

Recalling $x_1 = x_0$ and $\alpha_1 = \alpha_0$, we have $\frac{\alpha_1}{b_1} = \frac{\alpha_0}{b_0}$. Then, combining Eq.(23) and Eq.(29), we obtain the conclusion.

### A.2 PROOF OF THEOREM 1

**Proof.** By the the gradient Lipschitz continuous in Assumption 2 and Lemma 2, we obtain

$$\mathbb{E}[f(z_{t+1}) - f(z_t)] \leq \mathbb{E}\langle \nabla f(z_t), z_{t+1} - z_t \rangle + \frac{L}{2}\mathbb{E}\|z_{t+1} - z_t\|^2$$

$$= \mathbb{E}\left[\frac{\beta}{1-\beta}\langle \nabla f(z_t), \left(\frac{\alpha_{t-1}}{b_{t-1}} - \frac{\alpha_t}{b_t}\right) \odot m_{t-1}\rangle\right] - \mathbb{E}\left[\langle \nabla f(z_t), \alpha_t \frac{g_t}{b_t}\rangle\right]$$

$$- \mathbb{E}\left[\langle \nabla f(z_t), \alpha_{t-1}\delta_{t-1}\rangle\right] - \mathbb{E}\left[\langle \nabla f(z_t), (\alpha_t - \alpha_t)e_{t-1}\rangle\right]$$

$$+ \mathbb{E}\left[\frac{L}{2}\left\|\frac{\beta}{1-\beta}\left(\frac{\alpha_{t-1}}{b_{t-1}} - \frac{\alpha_t}{b_t}\right) \odot m_{t-1} - \alpha_t \frac{g_t}{b_t} - \alpha_{t-1}\delta_{t-1} - (\alpha_t - \alpha_{t-1})e_{t-1}\right\|^2\right]$$

$$= \mathbb{E}\left[\frac{\beta}{1-\beta}\langle \nabla f(z_t), \left(\frac{\alpha_{t-1}}{b_{t-1}} - \frac{\alpha_t}{b_t}\right) \odot m_{t-1}\rangle\right] - \mathbb{E}\left[\langle \nabla f(z_t), \alpha_t \frac{g_t}{b_t}\rangle\right]$$

$$+ \mathbb{E}\left[\frac{L}{2}\left\|\frac{\beta}{1-\beta}\left(\frac{\alpha_{t-1}}{b_{t-1}} - \frac{\alpha_t}{b_t}\right) \odot m_{t-1} - \alpha_t \frac{g_t}{b_t} - \alpha_{t-1}\delta_{t-1} - (\alpha_t - \alpha_{t-1})e_{t-1}\right\|^2\right]$$

$$\leq \mathbb{E}\left[\frac{\beta}{1-\beta}\langle \nabla f(z_t), \left(\frac{\alpha_{t-1}}{b_{t-1}} - \frac{\alpha_t}{b_t}\right) \odot m_{t-1}\rangle\right] - \mathbb{E}\left[\langle \nabla f(z_t), \alpha_t \frac{g_t}{b_t}\rangle\right]$$

$$+ L\mathbb{E}\left[\left\|\frac{\beta}{1-\beta}\left(\frac{\alpha_{t-1}}{b_{t-1}} - \frac{\alpha_t}{b_t}\right) \odot m_{t-1}\right\|^2\right] + L\mathbb{E}\left[\alpha_t^2 \left\|\frac{g_t}{b_t}\right\|^2\right]$$

$$+ \frac{L}{2}\mathbb{E}\left[\|\alpha_{t-1}\delta_{t-1}\|^2\right] + \frac{L}{2}\mathbb{E}\left[\|(\alpha_{t-1} - \alpha_t)e_t\|^2\right] \tag{30}$$

where the second equality holds due to $\mathbb{E}[\delta_{t-1}] = 0$ and $\mathbb{E}[e_{t-1}] = 0$. The last inequality holds owing to $\mathbb{E}[\|a+b\|^2] = \mathbb{E}[\|a\|^2] + \mathbb{E}[\|b\|^2]$ if $\mathbb{E}[a] = 0$ or $\mathbb{E}[b] = 0$, and $\mathbb{E}[\|a+b\|^2] \leq 2\mathbb{E}[\|a\|^2] + 2\mathbb{E}[\|b\|^2]$ if $\mathbb{E}[a] \neq 0$ and $\mathbb{E}[b] \neq 0$.

Taking telescope sum from 1 to $T$ on the both sides of Eq.(30) , we then have

$$\mathbb{E}[f(z_T) - f(z_1)] \leq \underbrace{\frac{\beta}{1-\beta}\mathbb{E}\left[\sum_{t=1}^{T}\langle \nabla f(z_t), \left(\frac{\alpha_{t-1}}{b_{t-1}} - \frac{\alpha_t}{b_t}\right) \odot m_{t-1}\rangle\right]}_{T_1} \underbrace{- \mathbb{E}\left[\sum_{t=1}^{T}\langle \nabla f(z_t), \alpha_t \frac{g_t}{b_t}\rangle\right]}_{T_2}$$

$$+ \underbrace{L\mathbb{E}\left[\sum_{t=1}^{T}\left\|\frac{\beta}{1-\beta}\left(\frac{\alpha_{t-1}}{b_{t-1}} - \frac{\alpha_t}{b_t}\right) \odot m_{t-1}\right\|^2\right]}_{T_3}$$

$$+ \underbrace{L\mathbb{E}\left[\sum_{t=1}^{T}\alpha_t^2 \left\|\frac{g_t}{b_t}\right\|^2\right] + \frac{L}{2}\mathbb{E}[\sum_{t=1}^{T}\|\alpha_{t-1}\delta_{t-1}\|^2] + \frac{L}{2}\mathbb{E}[\sum_{t=1}^{T}\|(\alpha_{t-1} - \alpha_t)e_t\|^2]}_{T_4}$$

$$\tag{31}$$

Now we focus on bounding $T_1$ below. From Assumption 4, we know $\|g_t\| \leq G$ $(t = 1, 2, ..., T)$ and $\|\nabla f(z_t)\| \leq G$ . Due to $m_t = \beta m_{t-1} + (1 - \beta)g_t$ and $m_1 = g_1$, it is easy to obtain $\|m_t\| \leq G$ by complete induction.

Since $\|\nabla f(z_t)\| \leq G$ and $\|m_t\| \leq G$, we have

$$
\begin{aligned}
T_1 &= \frac{\beta}{1 - \beta} \mathbb{E}\left[\sum_{i=1}^{T} \langle \nabla f(z_i), \left(\frac{\alpha_{t-1}}{b_{t-1}} - \frac{\alpha_t}{b_t}\right) \odot m_{i-1} \rangle\right] \\
&\overset{(i)}{\leq} \frac{\beta}{1 - \beta} \mathbb{E}\left[\sum_{i=1}^{T} \|\nabla f(z_t)\| \|m_t\| \left\|\frac{\alpha_{t-1}}{b_{t-1}} - \frac{\alpha_t}{b_t}\right\|_1\right] \\
&\overset{(ii)}{\leq} \frac{\beta}{1 - \beta} G^2 \mathbb{E}\left[\sum_{i=1}^{T} \left\|\frac{\alpha_{t-1}}{b_{t-1}} - \frac{\alpha_t}{b_t}\right\|_1\right] \\
&\overset{(iii)}{=} \frac{\beta}{1 - \beta} G^2 \mathbb{E}\left[\left\|\sum_{i=1}^{T} \left(\frac{\alpha_{t-1}}{b_{t-1}} - \frac{\alpha_t}{b_t}\right)\right\|_1\right] \\
&\leq \frac{\beta}{1 - \beta} G^2 \mathbb{E}\left[\left\|\frac{\alpha_0}{b_0}\right\|_1\right] \\
&\overset{(iv)}{\leq} \frac{\alpha_0 \beta d}{(1 - \beta)\rho} G^2,
\end{aligned}
\tag{32}
$$

where $(i)$ holds sice $\|a \odot b\| \leq \|a\| \max_j |(b)_j| \leq \|a\| \|b\|_1$, $(ii)$ holds due to $\|\nabla f(z_t)\| \leq G$ and $\|m_t\| \leq G$, $(iii)$ holds because $\frac{\alpha_{t-1}}{(b_{t-1})_j} - \frac{\alpha_t}{(b_t)_j} \geq 0$ for any $j \in [1, 2, ..., d]$, $(iv)$ holds due to $\min_j (b_t)_j \geq \rho > 0$ for any $j \in [1, 2, ..., d]$.

Let us turn to bound $T_2$,

$$
\begin{aligned}
T_2 &= -\mathbb{E}\left[\sum_{t=1}^{T} \langle \nabla f(z_t), \alpha_t \frac{g_t}{b_t} \rangle\right] \\
&= \underbrace{-\mathbb{E}\left[\sum_{t=1}^{T} \langle \nabla f(z_t) - f(x_t), \alpha_t \frac{g_t}{b_t} \rangle\right]}_{T_5} \underbrace{-\mathbb{E}\left[\sum_{t=1}^{T} \langle \nabla f(x_t), \alpha_t \frac{g_t}{b_t} \rangle\right]}_{T_6}
\end{aligned}
\tag{33}
$$

We now analyze $T_5$ below,

$$T_5 = -\mathbb{E}\left[\sum_{t=1}^{T}\langle\nabla f(z_t) - f(x_t), \alpha_t\frac{g_t}{b_t}\rangle\right]$$

$$\overset{(i)}{\leq}\frac{1}{2}\mathbb{E}\left[\sum_{t=1}^{T}\|f(z_t) - f(x_t)\|^2\right] + \frac{1}{2}\mathbb{E}\left[\sum_{t=1}^{T}\alpha_t^2\left\|\frac{g_t}{b_t}\right\|^2\right]$$

$$\overset{(ii)}{\leq}\frac{L^2}{2}\mathbb{E}\left[\sum_{t=1}^{T}\|z_t - x_t\|^2\right] + \frac{1}{2}\mathbb{E}\left[\sum_{t=1}^{T}\alpha_t^2\left\|\frac{g_t}{b_t}\right\|^2\right]$$

$$\overset{(iii)}{=}\frac{L^2}{2}\mathbb{E}\left[\sum_{t=1}^{T}\left\|\frac{\beta}{1-\beta}(x_t - x_{t-1}) + \frac{\alpha_{t-1}}{1-\beta}\left(\delta_{t-1} + \beta e_{t-1} - e_t\right)\right\|^2\right] + \frac{1}{2}\mathbb{E}\left[\sum_{t=1}^{T}\alpha_t^2\left\|\frac{g_t}{b_t}\right\|^2\right]$$

$$\overset{(iv)}{\leq}\frac{\beta^2 L^2}{(1-\beta)^2}\mathbb{E}\left[\sum_{t=1}^{T}\|x_t - x_{t-1}\|^2\right] + \frac{L^2}{(1-\beta)^2}\mathbb{E}\left[\sum_{t=1}^{T}\|\alpha_{t-1}\delta_{t-1}\|^2\right]$$
$$+ \frac{\beta^2 L^2}{(1-\beta)^2}\mathbb{E}\left[\sum_{t=1}^{T}\|\alpha_{t-1}e_{t-1}\|^2\right] + \frac{L^2}{(1-\beta)^2}\mathbb{E}\left[\sum_{t=1}^{T}\|\alpha_{t-1}e_t\|^2\right] + \frac{1}{2}\mathbb{E}\left[\sum_{t=1}^{T}\alpha_t^2\left\|\frac{g_t}{b_t}\right\|^2\right]$$

$$\overset{(v)}{=}\frac{\beta^2 L^2}{(1-\beta)^2}\mathbb{E}\left[\sum_{t=1}^{T}\alpha_{t-1}^2\left\|\left(\frac{m_{t-1}}{b_{t-1}} + \delta_{t-1} + e_{t-1} - e_t\right)\right\|^2\right] + \frac{L^2}{(1-\beta)^2}\mathbb{E}\left[\sum_{t=1}^{T}\|\alpha_{t-1}\delta_{t-1}\|^2\right]$$
$$+ \frac{\beta^2 L^2}{(1-\beta)^2}\mathbb{E}\left[\sum_{t=1}^{T}\|\alpha_{t-1}e_{t-1}\|^2\right] + \frac{L^2}{(1-\beta)^2}\mathbb{E}\left[\sum_{t=1}^{T}\|\alpha_{t-1}e_t\|^2\right] + \frac{1}{2}\mathbb{E}\left[\sum_{t=1}^{T}\alpha_t^2\left\|\frac{g_t}{b_t}\right\|^2\right]$$

$$=\frac{\beta^2 L^2}{(1-\beta)^2}\mathbb{E}\left[\sum_{t=1}^{T}\left\|\frac{\alpha_{t-1}m_{t-1}}{b_{t-1}}\right\|^2\right] + \frac{\beta^2 L^2}{(1-\beta)^2}\mathbb{E}\left[\sum_{t=1}^{T}\|\alpha_{t-1}\delta_{t-1}\|^2\right]$$
$$+ \frac{\beta^2 L^2}{(1-\beta)^2}\mathbb{E}\left[\sum_{t=1}^{T}\|\alpha_{t-1}e_{t-1}\|^2\right] + \frac{\beta^2 L^2}{(1-\beta)^2}\mathbb{E}\left[\sum_{t=1}^{T}\|\alpha_{t-1}e_t\|^2\right] + \frac{L^2}{(1-\beta)^2}\mathbb{E}\left[\sum_{t=1}^{T}\|\alpha_{t-1}\delta_{t-1}\|^2\right]$$
$$+ \frac{\beta^2 L^2}{(1-\beta)^2}\mathbb{E}\left[\sum_{t=1}^{T}\|\alpha_{t-1}e_{t-1}\|^2\right] + \frac{L^2}{(1-\beta)^2}\mathbb{E}\left[\sum_{t=1}^{T}\|\alpha_{t-1}e_t\|^2\right] + \frac{1}{2}\mathbb{E}\left[\sum_{t=1}^{T}\alpha_t^2\left\|\frac{g_t}{b_t}\right\|^2\right]$$

$$=\frac{\beta^2 L^2}{(1-\beta)^2}\mathbb{E}\left[\sum_{t=1}^{T}\alpha_{t-1}^2\left\|\frac{m_{t-1}}{b_{t-1}}\right\|^2\right] + \frac{(1+\beta^2)L^2}{(1-\beta)^2}\mathbb{E}\left[\sum_{t=1}^{T}\alpha_{t-1}^2\|\delta_{t-1}\|^2\right]$$
$$+ \frac{2\beta^2 L^2}{(1-\beta)^2}\mathbb{E}\left[\sum_{t=1}^{T}\alpha_{t-1}^2\|e_{t-1}\|^2\right] + \frac{(1+\beta^2)L^2}{(1-\beta)^2}\mathbb{E}\left[\sum_{t=1}^{T}\alpha_{t-1}^2\|e_t\|^2\right] + \frac{1}{2}\mathbb{E}\left[\sum_{t=1}^{T}\alpha_t^2\left\|\frac{g_t}{b_t}\right\|^2\right]$$

$$\overset{(vi)}{\leq}\left(\frac{\beta^2 L^2 d}{(1-\beta)^2} + \frac{4(1+\beta^2)L^2 d}{(1-\beta)^2} + \frac{2\beta^2 L^2 d}{(1-\beta)^2} + \frac{(1+\beta^2)L^2 d}{(1-\beta)^2} + \frac{G^2}{2\rho^2}\right)\sum_{t=1}^{T}\alpha_{t-1}^2$$

$$=\left(\frac{(8\beta^2 + 10\beta + 5)L^2 d}{(1-\beta)^2} + \frac{G^2}{2\rho^2}\right)\sum_{t=1}^{T}\alpha_{t-1}^2$$

$$(34)$$

where $(i)$ holds by following $\langle a, b\rangle \leq \frac{1}{2}\|a\|^2 + \frac{1}{2}\|a\|^2$, $(ii)$ holds due to Assumption 1, $(iii)$ holds due to Assumption 1 owing to Eq.(21), $(iii)$ holds since $\mathbb{E}[\|a+b\|^2] = \mathbb{E}[\|a\|^2] + \mathbb{E}[\|b\|^2]$ if $\mathbb{E}[a] = 0$ or $\mathbb{E}[b] = 0$, $(v)$ holds resulting from the updating rule in Eq. (14), $(vi)$ holds due to $\left|\frac{(m_t)_j}{(b_t)_j}\right| \leq 1$, $|(\delta)_j| \leq 2$ (the definition of $\delta_t$ in Eq. (11) ), $\mathbb{E}[\|e_t\|^2] \leq d$ in Lemma 1, $\|g_t\| \leq G$ in Assumption 2 and $\min_j (b_t)_j \geq \rho > 0$.

We then bound $T_6$

$$
\begin{aligned}
T_6 &= -\mathbb{E}\left[\sum_{t=1}^{T}\langle\nabla f(x_t), \alpha_t\frac{g_t}{b_t}\rangle\right]\\
&= -\mathbb{E}\left[\sum_{t=1}^{T}\langle\nabla f(x_t), \alpha_t\frac{\nabla f(x_t)}{b_t}\rangle\right] - \mathbb{E}\left[\sum_{t=1}^{T}\langle\nabla f(x_t), \alpha_t\frac{g_t - \nabla f(x_t)}{b_t}\rangle\right]\\
&\overset{(i)}{\leq} -\frac{1}{G}\mathbb{E}\left[\sum_{t=1}^{T}\alpha_t\|\nabla f(x_t)\|^2\right] + \mathbb{E}\left[\sum_{t=1}^{T}\langle\nabla f(x_t), \alpha_t\frac{\nabla f(x_t) - g_t}{b_t}\rangle\right]\\
&= -\frac{1}{G}\mathbb{E}\left[\sum_{t=1}^{T}\alpha_t\|\nabla f(x_t)\|^2\right] + \mathbb{E}\left[\langle\nabla f(x_1), \alpha_1\frac{\nabla f(x_1) - g_1}{b_1}\rangle\right]\\
&\quad + \mathbb{E}\left[\sum_{t=2}^{T}\langle\nabla f(x_t), \nabla(f(x_t) - g_t)\odot\left(\frac{\alpha_t}{b_t} - \frac{\alpha_{t-1}}{b_{t-1}}\right)\rangle\right] + \mathbb{E}\left[\sum_{t=2}^{T}\langle\nabla f(x_t), \alpha_{t-1}\frac{\nabla f(x_t) - g_t}{b_{t-1}}\rangle\right]\\
&\overset{(ii)}{=} -\frac{1}{G}\mathbb{E}\left[\sum_{t=1}^{T}\alpha_t\|\nabla f(x_t)\|^2\right] + \mathbb{E}\left[\langle\nabla f(x_1), \alpha_1\frac{\nabla f(x_1) - g_1}{b_1}\rangle\right]\\
&\quad + \mathbb{E}\left[\sum_{t=2}^{T}\langle\nabla f(x_t), (\nabla f(x_t) - g_t)\odot\left(\frac{\alpha_t}{b_t} - \frac{\alpha_{t-1}}{b_{t-1}}\right)\rangle\right]\\
&\overset{(iii)}{\leq} -\frac{1}{G}\mathbb{E}\left[\sum_{t=1}^{T}\alpha_t\|\nabla f(x_t)\|^2\right] + \mathbb{E}\left[\|\nabla f(x_1)\|\|\nabla f(x_1) - g_1\|\left\|\frac{\alpha_1}{b_1}\right\|_1\right]\\
&\quad + \mathbb{E}\left[\sum_{t=2}^{T}\|\nabla f(x_t)\|\|\nabla f(x_t) - g_t\|\left\|\frac{\alpha_t}{b_t} - \frac{\alpha_{t-1}}{b_{t-1}}\right\|_1\right]\\
&\overset{(iv)}{\leq} -\frac{1}{G}\mathbb{E}\left[\sum_{t=1}^{T}\alpha_t\|\nabla f(x_t)\|^2\right] + 2G^2\mathbb{E}\left[\left\|\frac{\alpha_1}{b_1}\right\|_1 + \sum_{t=2}^{T}\left\|\frac{\alpha_t}{b_t} - \frac{\alpha_{t-1}}{b_{t-1}}\right\|_1\right]\\
&\overset{(v)}{=} -\frac{1}{G}\mathbb{E}\left[\sum_{t=1}^{T}\alpha_t\|\nabla f(x_t)\|^2\right] + 2G^2\mathbb{E}\left[\left\|\frac{\alpha_1}{b_1} + \sum_{t=2}^{T}\frac{\alpha_{t-1}}{b_{t-1}} - \frac{\alpha_t}{b_t}\right\|_1\right],\\
&= -\frac{1}{G}\mathbb{E}\left[\sum_{t=1}^{T}\alpha_t\|\nabla f(x_t)\|^2\right] + 4G^2\mathbb{E}\left[\left\|\frac{\alpha_1}{b_1}\right\|_1\right],\\
&\overset{(vi)}{\leq} -\frac{1}{G}\mathbb{E}\left[\sum_{t=1}^{T}\alpha_t\|\nabla f(x_t)\|^2\right] + \frac{4G^2\alpha_1 d}{\rho}
\end{aligned}
$$

$$(35)$$

where $(i)$ holds due to $\max_j(b_t)_j \leq \|b_t\| \leq G$ , $(ii)$ holds owing to $\mathbb{E}[\nabla f(x_t) - g_t] = 0$ in Assumption 2 and $g_t, b_{t-1}$ are independent, $(iii)$ holds sice $\|a\odot b\| \leq \|a\|\max_j|(b)_j| \leq \|a\|\|b\|_1$, $(iv)$ holds resulting from $\|\nabla f(x_t)\| \leq G$ and $\|\nabla f(x_t) - g_t\| \leq \|\nabla f(x_t)\| + \|g_t\| \leq 2G$, and $(v)$ holds because $\frac{\alpha_{t-1}}{(b_{t-1})_j} - \frac{\alpha_t}{(b_t)_j} \geq 0$ for any $j \in [1, 2, ..., d]$, $(vi)$ holds due to $\min_j(b_t)_j \geq \rho > 0$ for any $j \in [1, 2, ..., d]$.

Then, we pay attention to $T_3$,

$$
\begin{aligned}
T_3 &= L\mathbb{E}\left[\sum_{t=1}^{T}\left\|\frac{\beta}{1-\beta}\left(\frac{\alpha_{t-1}}{b_{t-1}}-\frac{\alpha_t}{b_t}\right)\odot m_{t-1}\right\|^2\right] \\
&\overset{(i)}{\leq} \frac{\beta^2 L}{(1-\beta)^2}\mathbb{E}\left[\sum_{t=1}^{T}\left\|\frac{\alpha_{t-1}}{b_{t-1}}-\frac{\alpha_t}{b_t}\right\|^2\|m_{t-1}\|^2\right] \\
&\overset{(ii)}{\leq} \frac{\beta^2 LG^2}{(1-\beta)^2}\mathbb{E}\left[\sum_{t=1}^{T}\left\|\frac{\alpha_{t-1}}{b_{t-1}}-\frac{\alpha_t}{b_t}\right\|^2\right] \\
&\overset{(iii)}{\leq} \frac{\beta^2 LG^2}{(1-\beta)^2}\mathbb{E}\left[\sum_{t=1}^{T}\max_j\left|\frac{\alpha_{t-1}}{(b_{t-1})_j}-\frac{\alpha_t}{(b_t)_j}\right|\left\|\frac{\alpha_{t-1}}{b_{t-1}}-\frac{\alpha_t}{b_t}\right\|_1\right] \\
&\overset{(iv)}{\leq} \frac{\alpha_0\beta^2 LG^2}{\rho(1-\beta)^2}\mathbb{E}\left[\sum_{t=1}^{T}\max_j\left(\frac{\alpha_{t-1}}{(b_{t-1})_j}\right)\left\|\frac{\alpha_{t-1}}{b_{t-1}}-\frac{\alpha_t}{b_t}\right\|_1\right] \\
&\overset{(v)}{\leq} \frac{\alpha_0\beta^2 LG^2}{\rho(1-\beta)^2}\mathbb{E}\left[\sum_{t=1}^{T}\left\|\frac{\alpha_{t-1}}{b_{t-1}}-\frac{\alpha_t}{b_t}\right\|_1\right] \\
&\overset{(vi)}{\leq} \frac{\alpha_0\beta^2 LG^2}{\rho(1-\beta)^2}\mathbb{E}\left[\sum_{t=1}^{T}\left\|\frac{\alpha_{t-1}}{b_{t-1}}\right\|_1-\left\|\frac{\alpha_t}{b_t}\right\|_1\right] \\
&\overset{(vii)}{\leq} \frac{\alpha_0\beta^2 LG^2}{\rho(1-\beta)^2}\mathbb{E}\left[\left\|\frac{\alpha_0}{b_0}\right\|_1-\left\|\frac{\alpha_T}{b_T}\right\|_1\right] \\
&\overset{(viii)}{\leq} \frac{\alpha_0^2\beta^2 LG^2 d}{\rho^2(1-\beta)^2},
\end{aligned}
\tag{36}
$$

where $(i)$ holds due to $\|a\odot b\|\leq\|a\|\|b\|$, $(ii)$ holds owing to $\|m_{t-1}\|\leq G$, $(ii)$ holds due to $\|a\|^2\leq\max_j|(a)_j|\|a\|_1$, $(iv)$ holds due to $\frac{\alpha_{t-1}}{(b_{t-1})_j}-\frac{\alpha_t}{(b_t)_j}\geq 0$ and $\frac{\alpha_t}{(b_t)_j}>0$ for any $j\in[1,2,...,d]$, $(v)$ holds resulting from $\min_j(b_t)_j\geq\rho>0$ for any $j$ and $\alpha_t$ is non-increasing, $(vi)$ holds resulting from $\frac{\alpha_{t-1}}{(b_{t-1})_j}-\frac{\alpha_t}{(b_t)_j}\geq 0$ for any $j\in[1,2,...,d]$, $(vii)$ holds due to telescoping sum, and $(viii)$ holds due to $\min_j(b_t)_j\geq\rho>0$ for any $j\in[1,2,...,d]$..

Now we turn attention to $T_4$,

$$
\begin{aligned}
T_4 &= L\mathbb{E}\left[\sum_{t=1}^{T}\alpha_t^2\left\|\frac{g_t}{b_t}\right\|^2\right]+\frac{L}{2}\mathbb{E}[\sum_{t=1}^{T}\|\alpha_{t-1}\delta_{t-1}\|^2]+\frac{L}{2}\mathbb{E}[\sum_{t=1}^{T}\|(\alpha_{t-1}-\alpha_t)e_t\|^2] \\
&\leq\left(L\frac{G^2}{\rho^2}+2dL\right)\sum_{t=1}^{T}\alpha_t^2+\frac{dL}{2}\sum_{t=1}^{T}(\alpha_{t-1}-\alpha_t)^2,
\end{aligned}
\tag{37}
$$

where the inequality holds owing to $\|m_{t-1}\|\leq G$ and $\min_j(b_t)_j\geq\rho>0$, $\|(\delta_{t-1})_j\|\leq 2$, and $\mathbb{E}[\|e_t\|^2]\leq d$.

Combining Eq.(31-37), we can obtain

$$
\begin{aligned}
\mathbb{E}[f(z_T)-f(z_1)]\leq &\frac{\alpha_0\beta d}{(1-\beta)\rho}G^2+\left(\frac{(8\beta^2+10\beta+5)L^2 d}{(1-\beta)^2}+\frac{G^2}{2\rho^2}\right)\sum_{t=1}^{T}\alpha_{t-1}^2 \\
&-\frac{1}{G}\mathbb{E}\left[\sum_{t=1}^{T}\alpha_t\|\nabla f(x_t)\|^2\right]+\frac{4G^2\alpha_1 d}{\rho}+\frac{\alpha_0^2\beta^2 LG^2 d}{\rho^2(1-\beta)^2} \\
&+\left(L\frac{G^2}{\rho^2}+2dL\right)\sum_{t=1}^{T}\alpha_t^2+\frac{dL}{2}\sum_{t=1}^{T}(\alpha_{t-1}-\alpha_t)^2.
\end{aligned}
\tag{38}
$$

Reformulating Eq.(38), we then have

$$
\frac{1}{G}\mathbb{E}\left[\sum_{t=1}^{T}\alpha_t\|\nabla f(x_t)\|^2\right] \leq \mathbb{E}[f(z_1) - f(z_T)]
$$

$$
+ \left(\frac{(8\beta^2 + 10\beta + 5)L^2 d}{(1-\beta)^2} + \frac{G^2(1+L)}{2\rho^2} + 2dL\right)\sum_{t=1}^{T}\alpha_{t-1}^2 \tag{39}
$$

$$
+ \frac{dL}{2}\sum_{t=1}^{T}(\alpha_{t-1} - \alpha_t)^2
$$

$$
+ \frac{\alpha_0\beta d}{(1-\beta)\rho}G^2 + \frac{4G^2\alpha_1 d}{\rho} + \frac{\alpha_0^2\beta^2 LG^2 d}{\rho^2(1-\beta)^2}
$$

It is known the learning rate saftifies $\alpha_t = \frac{c}{\sqrt{t}}, \forall t \geq 1$ and $\alpha_0 = \alpha_1 = c$. Utilizing non-increasing $\alpha_t$ and Cauchy-Schwarz inequality, we know $\mathbb{E}\left[\sum_{t=1}^{T}\alpha_t\|\nabla f(x_t)\|^2\right] \geq T\alpha_T\mathbb{E}\left[\frac{1}{T}\sum_{t=1}^{T}\|\nabla f(x_t)\|\right]^2 = \frac{\sqrt{T}}{c}\mathbb{E}\left[\frac{1}{T}\sum_{t=1}^{T}\|\nabla f(x_t)\|\right]^2$. $\sum_{t=1}^{T}\alpha_{t-1}^2 = \sum_{t=1}^{T}\frac{c^2}{t} \leq c^2(1 + \int_1^{T-1}\frac{1}{t}dt) \leq c^2(1 + \log T)$, and $\sum_{t=1}^{T}(\alpha_{t-1} - \alpha_t)^2 = \sum_{t=2}^{T}(\alpha_{t-1} - \alpha_t)^2 \leq \sum_{t=2}^{T}\frac{c^2}{4(t-1)^3} \leq \frac{c^2}{4}(1 + \int_1^{T-2}t^{-3}dt) = \frac{c^2}{4}(\frac{3}{2} - \frac{1}{2(T-2)}) \leq \frac{3c^2}{8}$, we further have

$$
\mathbb{E}\left[\frac{1}{T}\sum_{t=1}^{T}\|\nabla f(x_t)\|\right]^2 \leq \frac{C_1}{\sqrt{T}} + \frac{C_2(1 + \log T)}{\sqrt{T}}, \tag{40}
$$

where we define

$$
C_1 = cG\left(\mathbb{E}[f(z_1) - f^*] + \frac{3c^2 dL}{16} + \frac{\beta cdG^2}{(1-\beta)\rho} + \frac{4cdG^2}{\rho} + \frac{c^2\beta^2 LG^2 d}{\rho^2(1-\beta)^2}\right), \tag{41}
$$

$$
C_2 = c^3 G\left(\frac{(8\beta^2 + 10\beta + 5)L^2 d}{(1-\beta)^2} + \frac{G^2(1+L)}{2\rho^2} + 2dL\right). \tag{42}
$$

# B  UNQUANTIZED BINSGDM

---

**Algorithm 2.** SoftSignSGD

---
1: **Input**: model parameter $x_0, x_1$ , the momentum $m_0^{(i)} = 0$, $b_0^{(i)} = 0$, the exponential moving average factor $\beta$, the learning rate sequence $\{\alpha_t\}$
2: **for** $t = 1, ..., T$ **do**
3:     Randomly sample $\xi_t$ and compute the gradient: $g_t = \nabla f(x_t; \xi_t)$
4:     Update the momentum $m_t$: $m_t = \beta m_{t-1} + (1-\beta)g_t$
5:     Update the momentum $b_t$: $b_t = \beta b_{t-1} + (1-\beta)|g_t|$
6:     Update the model parameter $x_{t+1}$: $x_{t+1} = x_t - \alpha_t\frac{m_t}{b_t}$
7: **end for**

---

We refer to *BinSGDM* without quantization as *SoftSignSGD*. The implementation details for *SoftSignSGD* are shown in Algorithm 2. The gradients need to be aggregated among different workers before *SoftSignSGD* is performed, just like full-precision *SGD* and *Adam*. Compared to *Adam*, the first difference is that we utilize the exponential moving average of the absolute gradient, *i.e.*, $b_t = (1-\beta)b_{t-1} + |g_t|$, as the denominator of the updating amount for *SoftSignSGD* rather than conventionally adopt the squared root of the exponential moving average of the squared gradient, *i.e.*, $\sqrt{v_t} = \sqrt{(1-\beta_2)v_{t-1} + (1-\beta_2)g_t^2}$. Another difference is that the exponential moving factors for the nominator $m_t$ and the denominator $b_t$ are the same in *SoftSignSGD*. Both of the distinctions make each element of the updating amount in SoftSignSGD satisfies $-1 \leq (\frac{m_t}{b_t})_j \leq 1, \forall j \in [1, 2, ..., d]$.

## B.1  EXPERIMENTAL RESULTS FOR TRAINING VGG16

We assess the performances of *Adam*, *SoftSignSGD* and *BinSGDM* with VGG-16 on CIFAR100. We sample a set of 128 examples with the replacement for each batch. $\beta$ for *SoftSignSGD* and

*BinSGDM* is set to 0.95, and $\beta_1, \beta_2$ for *Adam* is set to 0.9, 0.999. The weight decay is uniformly set to 0.05. To simplify the tuning process and ensure fair comparisons, in each case, we start with the same learning rate of 0.005, divide the learning rate by 10 after 75 and 130 epochs, and finally terminate the procedure after 150 epochs. As visually illustrated in Figure 5, the convergence speed and the test accuracy of *SoftSignSGD* and *BinSGDM* are comparative to *Adam* for training VGG-16 on CIFAR100.

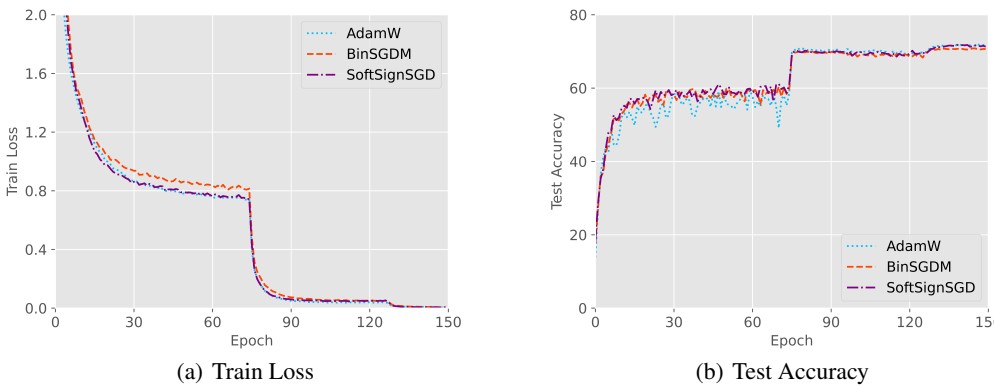

      (a) Train Loss                               (b) Test Accuracy

Figure 5: Training loss and test accuracy for VGG-16 on CIFAR100.

### B.2 EXPERIMENTAL RESULTS FOR TRAINING ViT

We train ViT-B with *Adam*, *SoftSignSGD* and *BinSGDM* on the ILSVRC2012. We use the Pytorch official implementation for ViT [5], and all experimental settings follow the recommended, expect that $\beta$ is to 0.95 for *SoftSignSGD* and *BinSGDM* and total epoch for all optimizers is set to 150 rather than 300. As shown in Figure 6, the convergence speed and the test accuracy of *SoftSignSGD* and *BinSGDM* can match *Adam* for training ViT-B-16 on ILSVRC2012.

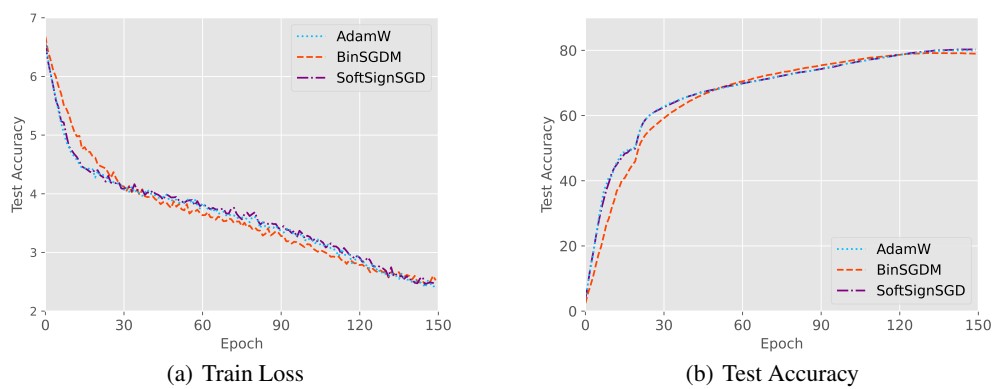

      (a) Train Loss                               (b) Test Accuracy

Figure 6: Training loss and test accuracy for ViT-B-16 on ILSVRC2012.

### B.3 EXPERIMENTAL RESULTS FOR TRAINING LSTM

We perform experiments for training a 3-layer LSTM on the Penn TreeBank dataset to validate the effectiveness of *SoftSignSGD*. Our implementations are built upon the codes of the paper AdaBelief [6], and we use the default experimental settings for adaptive optimizers in the code, expect that we set $\beta$ to 0.99 for *SoftSignSGD* and *BinSGDM* and the weight decay to 0.3 for all the optimizers. Figure 7 indicates that the convergence speed and the inference performance of *SoftSignSGD* and *BinSGDM* are competitive to the widely-used Adam.

---

[5]https://github.com/pytorch/vision/tree/main/references/classification
[6]https://github.com/juntang-zhuang/Adabelief-Optimizer

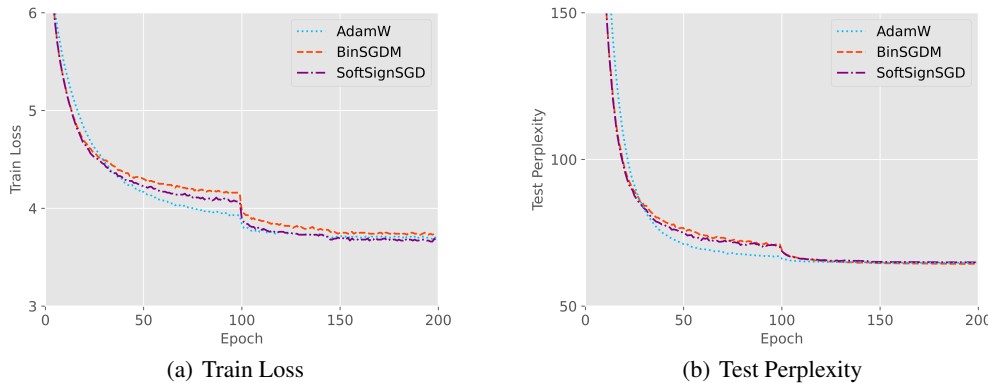

(a) Train Loss          (b) Test Perplexity

Figure 7: Training loss and test perplexity (the lower, the better) for 3-layer LSTM on Penn TreeBank.

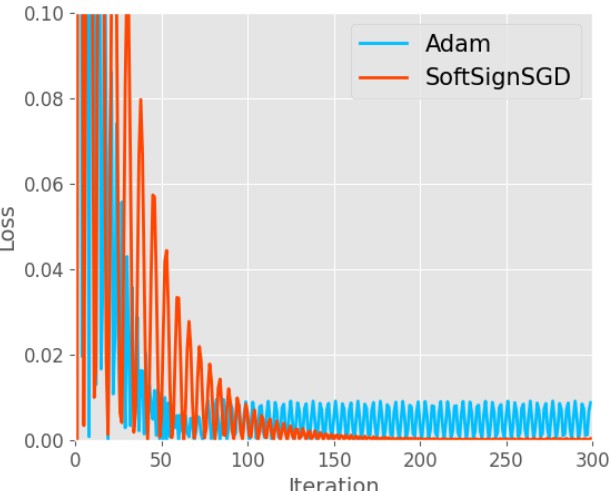

Figure 8: The convergence behaviors for *Adam* and *SoftSignSGD*. The loss function is $f(x) = 0.5x^2$. The learning rate is set to 0.5, $x_0$ is initialized to 1.0. $\beta_1$ and $\beta_2$ for *Adam* is set to 0.9 and 0.99, and $\beta$ for *SoftSignSGD* is set to 0.95.

## C  THE POTENTIAL OSCILLATION FOR *Adam*

Compared to *SoftSignSGD* (Please refer to Algorithm 2 in Section B), *Adam* may oscillate around the optimum and cannot approach it. As for *SoftSignSGD*, in the final optimizing stage, the elements of $g_t$ will frequently change their signs in the neighborhood of the optimal gradient $g^* = 0$. The more frequently the sign of the element of the gradient, $(g_t)_j, \forall j \in [1, 2, ..., d]$, change, the according $(m_t)_j$ will be smaller than the according $(b_t)_j$. Hence, when the loss approaches a local optimum, $\frac{m_t}{b_t}$ will be gradually close to 0. As for *Adam*, the update amount is $\frac{m_t}{\sqrt{v_t}}$ where $m_t = \beta_1 m_{t-1} + \beta_1 g_t$ and $v_t = \beta_2 v_t + (1 - \beta_2)v_t^2$. Although the elements of the gradient will frequently change their signs in the final optimizing stage, the relationship of size between $(m_t)_j$ and $(\sqrt{v_t})_j$ for any $j \in [1, 2, ..., d]$ is uncertain. Hence when the loss is close to a local optimum, the updating amount $\frac{m_t}{\sqrt{v_t}}$ may not damp to 0, which may lead the loss to be oscillating and not indeed converge to a local minimum. We provide an example in Figure 8 to visually illustrate the convergence behaviors for *Adam* and *SoftSignSGD*.

Table 3: System throughput (samples/s) of *AdamW*, *1-bit Adam* and *BinSGDM* for training ResNet-50 on ILSVRC2012 with 10Gbps Ethernet and 200Gbps InfiniBand.

| #GPUs | Optimizer | Ethernet (10Gbps) | | | InfiniBand (200Gbps) | | |
|---|---|---|---|---|---|---|---|
| | | Throughput (samples/s) | Speedup | Scale Efficiency | Throughput (samples/s) | Speedup | Scale Efficiency |
| 8 | SGDM | 3693 | 1.00× | 100% | 3693 | 1.00× | 100% |
| | 1-bit Adam | 3243 | 0.83× | 100% | 3243 | 0.83× | 100% |
| | BinSGDM | 3462 | 0.94× | 100% | 3462 | 0.94× | 100% |
| 16 | SGDM | 2959 | 1.00× | 40.1% | 4673 | 1.00× | 63.2% |
| | 1-bit Adam | 4715 | 1.60× | 72.7% | 5708 | 1.22× | 88.0% |
| | BinSGDM | 6015 | 2.03× | 86.9% | 6784 | 1.45× | 97.9% |
| 32 | SGDM | 4270 | 1.00× | 28.9% | 9063 | 1.00× | 61.3% |
| | 1-bit Adam | 7268 | 1.70× | 56.0% | 10249 | 1.13× | 79.0% |
| | BinSGDM | 9416 | 2.21× | 68.0% | 12131 | 1.34× | 87.6% |
| 32 | SGDM | 6189 | 1.00× | 20.9% | 16608 | 1.00× | 56.2% |
| | 1-bit Adam | 5546 | 0.89× | 21.3% | 16920 | 1.02× | 65.2% |
| | BinSGDM | 15253 | 2.47× | 55.1% | 19956 | 1.21× | 72.1% |

Table 4: System throughput (samples/s) of *AdamW*, *1-bit Adam* and *BinSGDM* for fine tuning BERT-Base on SQuAD 1.1 with 10Gbps Ethernet and 200Gbps InfiniBand.

| #GPUs | Optimizer | Ethernet (10Gbps) | | | InfiniBand (200Gbps) | | |
|---|---|---|---|---|---|---|---|
| | | Throughput (samples/s) | Speedup | Scale Efficiency | Throughput (samples/s) | Speedup | Scale Efficiency |
| 8 | AdamW | 413 | 1.00× | 100% | 413 | 1.00× | 100% |
| | 1-bit Adam | 358 | 0.87× | 100% | 358 | 0.83× | 100% |
| | BinSGDM | 412 | 1.00× | 100% | 412 | 0.94× | 100% |
| 16 | AdamW | 84 | 1.00× | 10.1% | 272 | 1.00× | 32.9% |
| | 1-bit Adam | 213 | 2.54× | 29.7% | 522 | 1.92× | 72.9% |
| | BinSGDM | 431 | 5.13× | 52.3% | 776 | 2.85× | 94.1% |
| 32 | AdamW | 119 | 1.00× | 7.20% | 543 | 1.00× | 32.8% |
| | 1-bit Adam | 274 | 2.30× | 19.1% | 903 | 1.66× | 63.1% |
| | BinSGDM | 730 | 6.13× | 44.2% | 1365 | 2.51× | 82.9% |
| 32 | AdamW | 158 | 1.00× | 4.78% | 998 | 1.00× | 30.2% |
| | 1-bit Adam | 252 | 1.59× | 8.80% | 1496 | 1.50× | 52.2% |
| | BinSGDM | 990 | 6.26× | 30.0% | 2299 | 2.30× | 69.8% |

## D  Experiments with InfiniBand connections

To further evaluate the communication efficiency of *SGDM/Adam*, *SoftSignSGD* and *BinSGDM* with high bandwidth connections, we implement experiments for training ResNet-50 and BERT-Base with distributed nodes connected with 200Gbps InfiniBand. All the experimental settings are the same as we perform experiments with Ethernet in Subsection 5.1, and the experimental results are listed in Table 3 and Table 4.

As shown in Table 3 and Table 4, compared with the baseline *SGDM/Adam*, *BinSGDM* can still reach up to $1.45\times$ speedup for ResNet-50 on ILSVRC2012 and $2.85\times$ speedup for BERT-Base on SQuAD 1.1, although the speed advantage is not so obvious as that with lower-bandwidth Ethernet connections. An interesting phenomenon is that the system throughput of *BinSGDM* with 10Gbps Ethernet can match that of *SGDM/Adam* with 200Gbps InfiniBand.

The experimental results in Table 3 and Table 4 also show that as the number of GPUs is increasing, the scale efficiency of *SGDM/Adam*, *SoftSignSGD* and *BinSGDM* becomes lower. The reason for this phenomenon can be summarized in the following. When the number of GPUs doubles, the number of communication trips also multiplies. We take the communication scheme *All-Reduce* for example. If the number of GPUs is $n$, each GPU requires $2(n-1)$ trips across the network confections. When the number is non-trivial, the computation time of the communication primitives may exceed the time of the pure communication itself and dominate the overall communication time, since the total communication overhead does not change with the number of GPUs. Notably, *All-reduce* is more efficient than *All-to-All* which is the core of our *Hierarchical 1-bit* . Hence, as shown in in Table 3 and Table 4, the scale efficiency of *BinSGDM* decreases more quickly than *SGDM/Adam* with the number of GPUs growing.

## E  Discussion

In the original *1-bit Adam* paper (Tang et al. (2021)), it reports that *1-bit Adam* runs significantly (up to $3.8\times$) faster than the full-precision *Adam*. Moreover, as the number of GPUs grows, the speed advantage is more obvious. In contrast, in our experiments, *1-bit Adam* does not exhibit clear speed advantages over the original *Adam*, and when running on 64 GPUs, *1-bit Adam* is not only slower than the original *Adam*, but also its throughput rate is even lower than that on 32 GPUs. The reason for this phenomenon can be summarized in the following. *First*, in (Tang et al. (2021)), the speedup of 1-bit *Adam* is obtained by comparing the throughput at the compression phase with the throughput at the warm-up phase. The warmup phase is excluded for assessing throughput, while, in our experiments, we evaluate the overall average throughput of the warm-up phase and the compression phase for *1-bit Adam*. *Second*, the baseline original *Adam* in (Tang et al. (2021)) does not run with system-level efficient *DDP*. *Third*, in (Tang et al. (2021)), the authors customized highly efficient communication primitives for *1-bit Adam*. For the sake of fairness, we just utilize the off-the-shelf communication primitives in Pytorch for all the optimizers.

As shown in Figure 4, when the number of GPUs continually, the communication time for *BinSGDM* also grows superlinearly. One of the reasons is that the communication primitive *All-to-All* accounts for more and more communication time. But the native *All-to-All* in Step $(iii)$ in *Hierarchical 1-bit All-Reduce* is not less efficient than the native *All-Reduce*. Hence, we will further optimize *All-to-All* and *All-Gather* to further accelerate *BinSGDM*.

When training large-scale DNNs, the mix-precision technique is used to reduce the memory, which allows us to further increase the model size. In optimizers, we still use the full-precision state and full-precision computation which commonly accounts for 33-75% of the total memory footprint (*Tim Dettmers, et al. 8-bit Optimizer via Block-wise Quantization, ICLR 2022.*). *BinSGDM* does not need full-precision state and full-precision computation. Moreover, due to randomly quantizing the update to $1$ or $-1$, *BinSGDM* may leverage lower precision than FP16 gradients to estimate the update. Therefore, *BinSGDM* is promising to find more applications in reducing memory.

