# OpenReview forum: "BinSGDM:  Extreme One-Bit Quantization for Communication Efficient Large-Scale Distributed Training "
_ICLR.cc/2023/Conference — Submitted to ICLR 2023_

### Official Review · Reviewer_YNiH · 2022-10-24

**Confidence:** 3
**Correctness:** 3
**Technical Novelty And Significance:** 3
**Empirical Novelty And Significance:** Not applicable
**Recommendation:** 6

**Clarity, Quality, Novelty And Reproducibility:**

The clarity of the paper is hampered a bit as the paper is riddled with writing issues. See above

**Strength And Weaknesses:**

Strength:
1. novel quantization scheme which is easy to implement
2. Provable convergence as update is same in expectation
3. strong empirical results


Weakness
1. writing needs a lot of work - see some confusions / point outs above


**Summary Of The Paper:**

The paper proposes to quantize ( two values)  the updates of the adam optimizer. The approach leverages the bounded nature of adam updates to design a unbiased quantization scheme - it uses the value of the update to define a bernoulli distribution (+1, -1). They show improvements in speed in several experiments.

Some questions /  comments,
1. What is \mathcal{M} on page two . It is not defined.
2. what does "random gradient of f" mean? Also i believe g_t is a vector . so what is g_t / |g_t|  .. Do you mean elementwise division?
3. Page 4, "Adam do not have this appealing property" can you explain this point further. The point you are making claims that adam does not converge at local minimas but keeps oscillating.
4. Theorem 1
      a. statement about "spectral norm of b_i" is bounded. Are we assuming b_i to be a matrix?
5. lot of grammatical errors in the paper. like neede, Lagre, "cost become a bottleneck",  "be friendly to efficient primitive communication primitives" and a lot more.

**Summary Of The Review:**

The paper seems to be making a novel contribution.

---

> ### Author Response · Authors · 2022-11-15
> **Response to YNiH**
>
> Heartfelt thanks for your encouraging and valuable comments. We are very glad to learn that you acknowledged our novel contributions; we believe that this work will contribute to the distributed learning. Further, we thank you for recognizing the provable theoretical analysis and the strong empirical results.  We agree with you that the original version had some grammatical errors and writing issues. We have carefully corrected them as well as similar issues. Below, we will explain your concerns and questions point by point.
>
> **Q1.**  $\mathcal{M}(\cdot)$ is not defined.
>
> **R1.** Thank you for checking so carefully. Yes, we forgot to define it.  We originally wanted to use $\mathcal{M}(\cdot)$ simply represents a momentum operator in the original manuscript, but we find it may be still unclear, so we substitute  $ {\mathcal Q}\left(\frac{{\mathcal M} _{\beta}(g_t)} {{\mathcal{M} _{\beta}(|g_t|)}}\right)$ with ${\cal Q}\left(\frac{m_t} {b_t}\right)$ where  $ m_t = \beta m _{t-1} + (1-\beta)g_t$ , $ b_t = \beta b _{t-1} + (1-\beta)|g_t|$  and $g_t$ is the gradient in the updated manuscript.
>
> **Q2.**  "The random gradient of $f(x_t)$"  and "$\frac{g_t} {|g_t|}$" are not clear.
>
> **R2.** We are sorry that  "the random gradient" made you confused, and we have rephrased it as "the estimated noisy gradient of $f(x_t)$ with random samples." Yes,  the divider here is an element-wise divider, and we have added this explanation in the revised manuscript.
>
>
> **Q3.** Further explain  "Adam do not have this appealing property" that may make Adam keep oscillating in the neighborhood of the local minima.
>
> **R3.** We first explain the appealing property in more detail. The update amount for uncompressed BinSGDM (we refer it to as SoftSignSGD) is $\frac{m_t}{{b_t}}$ where $m_t=\beta m_{t-1} + \beta g_t$ and $b_t = \beta b_t + (1-\beta)|g_t|$. In the final optimizing stage, the elements of $g_t$ will frequently change their signs in the neighborhood of the optimal gradient $g^*=0$. The more frequently the sign of a element of the gradient, $(g_t)_j, \forall j \in [1,2,...,d] $,  changes, the according $(m_t)_j$ will be smaller than the acoording $(b_t)_j$. Hence, when the loss approaches a local optimum, $\frac{m_t}{b_t}$ will be gradually close to $0$. As for Adam, the update amount is $\frac{m_t}{\sqrt{v_t}}$ where $m_t=\beta_1 m _{t-1} + \beta_1 g_t$ and $v_t = \beta_2 v _{t-1} + (1-\beta_2)v_t^2$. Although the elements of $g_t$ will frequently change their signs in the final optimizing stage, the relationship of size between  $(m_t)_j$ and $(\sqrt{v_t})_j$ for any $j \in [1,2, ..., d]$ is uncertain. Hence when the loss is close to a local optimum,  the update amount $\frac{m_t}{\sqrt{v_t}}$ may not damp to $0$, which may lead the loss to be oscillating and not indeed converge to a local minimum. We provide an example to visually illustrate this phenomenon in Appendix C of the updated version.
>
> **Q4.** The problem about $b_t^{(i})$  in Theorem 1.
>
> **R4.** Thank you for pointing out this less rigorous statement. The original statement $b_t^{(i)} \succeq \rho I$ is confusing and a little problematic, and $b_t^{(i})$ is a vector,  so we revise the statement as $(b_t^{(i)})_j \ge \rho >0 $, $\forall j \in [1,2,...,d] $.
>
>
> **Q5.**  Grammatical errors in the paper.
>
> **R5.** We were in a hurry to meet the deadline and did not have time to double-check the original manuscript, so there are some grammatical errors and typos in the original manuscript.  We have proofread  the manuscript and corrected  all the writing issues and typos you pointed out and other similar writing issues. Thank you for your precious comments that definitely have  helped us to enhance the quality of the manuscript.

---

> > ### Author Response · Authors · 2022-12-10
> > **Sincerely looking forward post-rebuttal feedback (the final discussion deadline is due in one day)**
> >
> > Dear Reviewer YNiH,
> >
> > We are sorry to bother you again.  We highly appreciate your positive comment that our paper makes a novel contribution. Your main concerns are about the writing issues. We had fixed all these writing issues you pointed out and other similar problems, and we had double checked and polished the manuscript to enhance the writing quality.  Moreover, we had given a detailed explanation *why  Adam may oscillate in the neighborhood of the local minima* you were concerned about  in the previous responses and a visual example in the revised manuscript.
> >
> > Since our paper is at the borderline , we really need your support. If we have addressed all your concerns, it is kind of you to increase the score.
> >
> > Thanks,
> >
> > authors

---

### Official Review · Reviewer_C7p2 · 2022-10-26

**Confidence:** 3
**Clarity, Quality, Novelty And Reproducibility:** The paper is well written and easy to…
**Correctness:** 3
**Technical Novelty And Significance:** 3
**Empirical Novelty And Significance:** 2
**Recommendation:** 5

**Strength And Weaknesses:**

There are four key contributions listed by the authors

1. The first algorithm that quantizes the entire update of an adaptive optimizer (which seems to me that it translates to less quantization steps) and does not need warm-up.
  - This one looks novel and simplifies the usage of this algorithm.
2. BinSGDM has the same convergence rate as the full-precision Adam
  -  This one is based on strong assumptions such as the gradients are bounded by a constant, and it doesn't say much since the convergence rate is also the same as SGD's 1/\sqrt{T} rate. A simple quantized SGD can achieve the same rate.
3. Hierarchical 1-bit All-Reduce
  - This is a standard system implementation, basically it treats a node as a single worker in distributed training. Already implemented in frameworks such as DDP, Horovod, Bagua, etc.
4. First work to be consistently faster than DDP
  - This is highly dependent on the system implementation and computer network. And also not true, Existing works like Bagua and HiPress already show significant speedup over DDP

Therefore the novelty seems to be 1, but since the algorithm is completely different from the traditional Adam (the second-moment stat is also replaced by first-moment of the gradient absolute value), more comprehensive experiments are needed to confirm the uncompressed version of the algorithm matches the convergence of Adam on most tasks.

Also since the paper claims to have speedup in real-world settings, it should use more practical settings for experiments. Most people don't use A100-80G GPUs with only 10Gbps ethernet connection.

**Summary Of The Paper:**

The authors proposed a new quantized distributed stochastic gradient algorithm which resembles Adam.

They prove the new algorithm BinSGDM bears the same convergence rate as Adam, and BinSGDM is more communication efficient than Adam. They also compare the new algorithm with other algorithms, including ADAMW, 1-bit Adam and SGDM to show that the BinSGDM implementation is more performant than other algorithms' implementations.

**Summary Of The Review:**

The algorithm looks novel, but there remains some work to justify the effectiveness of the algorithm.

1. The experiments need to be done in more realistic settings (for example A100 with RDMA connections)
2. Since the algorithm is based on a different update rule than the original Adam, to justify the new algorithm is as good as Adam (as one of the key contributions states) more comprehensive experiments are needed to confirm the uncompressed version of the algorithm matches the convergence of Adam on most tasks.

Also some claims need to be adjusted according to the "Strength And Weaknesses" section.

---

> ### Author Response · Authors · 2022-11-15
> **Response to C7p2 Part 1/3**
>
> We greatly appreciate your professional and constructive comments. We are pleased that you acknowledged the novelty of the algorithm itself. Also, we thank you for kindly suggesting us to add more extensive experiments to demonstrate the effectiveness of the proposed method, which is exceedingly helpful in improving the convincingness of the paper. Next, we will explain your concerns point by point in the following.
>
>
> **Q1.** The experiments need to be done in more settings.
>
> **R1.** In our laboratory, the highest Ethernet bandwidth   among the nodes is 10Gbps. During the preparation of the original manuscript, the InfiniBand service was not available, although we had physical InfiniBand connections. In these days, we urged the supplier to help us to manage it. As you suggested, we have implemented the experiments  on \{2, 4, 8\} nodes with 200Gbps InfiniBand, and we report the experimental results in the following.
>
> **Table 1.** system throughput  for training ResNet-50 on ILSVRC2012
> |#GPUs | Optimizer|Ethernet (10Gbps)|Ethernet (10Gbps)|Ethernet (10Gbps) |InfiniBand (200Gbps)| InfiniBand (200Gbps)| InfiniBand (200Gbps)|
> |:---------:|:---------|:------:|:---------:|:---------:|:---------:|:---------:|:---------:|
> |           |           | Throughput (samples/s)| Speedup| Scale Efficiency|Throughput (samples/s)| Speedup| Scale Efficiency|
> | 8        |    SGDM     |  3693|  1.00$\times$|   100%    |3693|  1.00$\times$|   100%|
> | 8        |   1-bitAdam| 3243|  0.83$\times$ |   100%   |3243|   0.83$\times$|   100%|
> | 8        |   BinSGDM | 3462|  0.94$\times$ |   100%   |3462|  0.94$\times$|   100%|
> | 16      |    SGDM     | 2959|  1.00$\times$ |   40.1%  |4673|  1.00$\times$|    63.2%|
> | 16      |   1-bitAdam| 4715|  1.60$\times$ |  72.7%   |5708|  1.22$\times$|   88.0%|
> | 16      |   BinSGDM |6015|   2.03$\times$| 86.9%     |6784|   1.45$\times$|   97.9%|
> | 32      |    SGDM     | 4270|  1.00$\times$ |   28.9%  |9063|  1.00$\times$|    61.3%|
> | 32      |   1-bitAdam| 7268|  1.70$\times$ |  56.0%   |10249| 1.13$\times$|   79.0%|
> | 32      |   BinSGDM |9416|   2.21$\times$| 68.0%     |12131|   1.34$\times$|   87.6%|
> | 64      |    SGDM     | 6189|  1.00$\times$ |   20.9%   |16608| 1.00$\times$|    56.2%|
> | 64      |   1-bitAdam| 5546|  0.89$\times$ |  21.3%   |16920| 1.02$\times$|   65.2%|
> | 64      |   BinSGDM |15253|   2.47$\times$| 55.1%   |19956|  1.21$\times$|   72.1%|
>
> **Table 2.** system throughput  for training BERT-Base on  SQuAD 1.1
> |#GPUs | Optimizer| Ethernet (10Gbps)|Ethernet (10Gbps)|Ethernet (10Gbps) | InfiniBand (200Gbps) | InfiniBand (200Gbps)| InfiniBand (200Gbps)|
> |:---------:|:---------|:---------:|:---------:|:---------:|:---------:|:---------:|:---------:|
> |           |           | Throughput (samples/s)| Speedup| Scale Efficiency|Throughput (samples/s)| Speedup| Scale Efficiency|
> | 8        |    AdamM    |  413|  1.00$\times$|   100%    |413|  1.00$\times$|   100%|
> | 8        |   1-bitAdam|  358|  0.87$\times$ |   100%   |358|   0.87$\times$|   100%|
> | 8        |   BinSGDM |  412|  1.00$\times$ |   100%   |412|   1.00$\times$|   100%|
> | 16      |    AdamM     | 84 |  1.00$\times$ |   10.1%  |272|  1.00$\times$|    32.8%|
> | 16      |   1-bitAdam| 213|  2.54$\times$ |  29.7%   |522|   1.92$\times$|   72.9%|
> | 16      |   BinSGDM | 431|   5.13$\times$|  52.3%    |776|   2.85$\times$|   94.1%|
> | 32      |    AdamM     | 119|  1.00$\times$ |   7.20%  |543|  1.00$\times$|    32.3%|
> | 32      |   1-bitAdam| 274|  2.30$\times$ |  19.1%   |903| 1.66$\times$|   63.1%|
> | 32      |   BinSGDM |730|   6.13$\times$|   44.2%    |1365|   2.51$\times$|   82.9%|
> | 64      |    AdamM     | 158|  1.00$\times$ |  4.78%   |998| 1.00$\times$|    30.2%|
> | 64      |   1-bitAdam| 252|  1.59$\times$ |  8.80%   |1496| 1.50$\times$|   52.2%|
> | 64      |   BinSGDM | 990|   6.26$\times$| 30.0%   |2299|  2.30$\times$|   69.8%|
>
> When we connect nodes with 200Gbps InfiniBand, BinSGDM can still achieve at least $1.2\times$ speedup for ResNet-50 and at least $2.3\times$ speedup for Bert-Base, comparing to the widely-used full-precision SGDM/AdamW. Another interesting phenomenon is that the system throughput of BinSGDM with 10Gbps Ethernet can match that of SGDM/Adam with 200Gbps InfiniBand. For more analysis please refer to Section C in the Appendix of the reivised manuscript.

---

> > ### Author Response · Authors · 2022-11-15
> > **Response to C7p2 Pat 2/3**
> >
> > **Q2.** More experiments are needed to confirm the uncompressed version of the algorithm matches the convergence of Adam on most tasks.
> >
> > **R2.** Thank you for this kind advice. As you suggested, we implement more experiments for training VGG16 on Cifar100,  ViT-B16 on ImageNet and LSTM on Penn TreeBank to validate the effectiveness of **the uncompressed version of BinSGDM, referred to as SoftSignSGD**. The experimental results are in presented below.
> >
> > **Table 3.** Train Loss and test accuracy  of AdamW, SoftSignSGD and BinSGDM for training VGG16 on CIFAR100
> > |Optimizers | Metric  | Epoch #20 | Epoch #40 | Epoch #60 | Epoch #80 | Epoch #100 | Epoch #120 | Epoch #140 | Epoch #150|
> > | :--------------|:--------------------| :------------: |:--------------: |:-------------: |:-------------: |:----------------: |:----------------: |:----------------: |:----------------: |
> > |Adam         |Train&ensp;Loss| 1.01    |0.81   |0.76  |0.14  |0.039  |0.042  |0.0071  |0.0043|
> > |SoftSignSGD|Train&ensp;Loss| 0.97   |0.82   |0.77  |0.14  |0.050  |0.049   |0.0063  |0.0045|
> > |BinSGDM      |Train&ensp;Loss| 1.05   |0.88   |0.83  |0.18  |0.058  |0.051  | 0.0088  |0.0062|
> > |AdamW|        Test&ensp;Acc  |55.08   |54.96   |59.41  |70.79  |69.50  |69.38  |71.50  |71.79|
> > |SoftSignSGD |Test&ensp;Acc| 57.40  |56.35   |59.85  |69.75  |69.39  |69.05  |71.69  |71.77|
> > |BinSGDM       |Test&ensp;Acc| 58.21  |57.29   |59.22  |69.76  |69.42  |69.02  |70.68  |70.82 |
> >
> > **Tabel 4.** Train Loss and test accuracy  of AdamW, SoftSignSGD and BinSGDM for training ViT-B16 on ImageNet
> > |Optimizers | Metric  | Epoch #20 | Epoch #40 | Epoch #60 | Epoch #80 | Epoch #100 | Epoch #120 | Epoch #140 | Epoch #150|
> > | :--------------|:--------------------| :------------: |:--------------: |:-------------: |:-------------: |:----------------: |:----------------: |:----------------: |:----------------: |
> > |Adam         | Train Loss| 4.27   |4.07   |3.74  |3.60  |3.21  |2.93  |2.52  |2.44
> > |SoftSignSGD|Train Loss| 4.41  |4.02   |3.79  |3.65  |3.24    |2.94     |2.53  |2.45
> > |BinSGDM      |Train Loss | 4.62  |3.94  |3.67  |3.40  |3.09  |2.81 |2.59 |2.52
> > |AdamW|        Test Acc  |50.12   |65.73   |69.54  |72.72  |75.75  |78.49  |79.99  |79.96
> > |SoftSignSGD |Test Acc  | 49.81  |65.76  |69.67 |72.86  |75.79  |78.51  |80.16 |80.24|
> > |BinSGDM       |Test Acc  |45.90  |64.02  |70.32  |74.79 |76.52  |78.62 |79.12 |79.97
> >
> > **Tabel 5.**  Train Loss and test perplexity (**the lower,   the better**)  of AdamW, SoftSignSGD and BinSGDM for training a 3-layer LSTM on Penn TreeBank
> > |Optimizers | Metric  | Epoch #10 | Epoch #30 | Epoch #60 | Epoch #90 | Epoch #120 | Epoch #150 | Epoch #180 | Epoch #200
> > | :--------------|:--------------------| :------------: |:--------------: |:-------------: |:-------------: |:----------------: |:----------------: |:----------------: |:----------------: |
> > |Adam         | Train Loss|  5.53 |4.48   |4.08   |3.95  |3.76  |3.71  |3.71  |3.69
> > |SoftSignSGD|Train Loss| 5.34|4.45   |4.17   |4.08  |3.76  |3.69  |3.68  |3.66
> > |BinSGDM      |Train Loss|5.35 |4.49   |4.25   |4.17  |3.84  |3.75  |3.74  |3.73
> > |AdamW|        Test Perplexity $\downarrow$|207.63   |84.44   |69.46  |67.22  |65.11  |64.81  |64.77  |64.76
> > |SoftSignSGD |Test Perplexity $\downarrow$|172.85  |84.01   |72.45  |70.31  |65.67  |65.08  |64.94  |64.93
> > |BinSGDM       |Test Perplexity $\downarrow$|173.33  |85.36  |74.26  |71.36  |65.56  |64.63  |64.47  |64.41
> >
> >
> > As shown in Tables 3-5, the experimental results indicate that the convergence speed and the inference performance of SoftSignSGD and BinSGDM can match that of the widely used Adam. For more implementation details and visual figures,  please refer to Section B in the Appendix of the updated manuscript.

---

> > > ### Author Response · Authors · 2022-11-17
> > > **Response to C7p2 Part 3/3**
> > >
> > > **Q3.** The convergence analysis is based on strong assumptions such as the gradients are bounded by a constant, and the convergence rate is also the same as SGD's $1/\sqrt{T}$ rate.
> > >
> > > **R3.**  We respectfully disagree with you on this comment. The assumption that the gradients are bounded by a constant might be a bit strong, but it is still commonly satisfied for training DNNs in practice. To the best of our knowledge , the theoretical convergence demonstration for all existing  Adam-type optimizers  needs this assumption, and their proven optimal convergence rates are also $O(1/\sqrt{T})$. It is widely accepted that the theoretical convergence analysis in these works  is one of their contributions. Just like these works, our paper is also not a pure theory-oriented work, and we do not pursue theoretically faster convergence rate or theoretically weaker assumptions. We think demonstrating a new optimizer  working well in practice achieves the known optimal convergence can still give new insights.  Hence, we believe that Theorem 1 that shows 1-bit BinSGDM can achieve the same convergence rate as the full-precision Adam with the same assumptions is theoretically meaningful.
> > >
> > >
> > > **Q4.**  *Hierarchical 1-bit All-Reduce* is a standard system implementation in frameworks such as DDP, Horovod, Bagua, etc.
> > >
> > > **R4.**  There are still some differences between *Hierarchical 1-bit All-Reduce* and the standard (Hierarchical) *All-Reduce* in these frameworks. Unlike the standard hierarchical implementation in frameworks, such as Bagua, we do not first aggregate tensors over local workers inside each node to a leader worker, and then perform inter-node aggregation over leader workers of different nodes,  and finally make each leader worker broadcast aggregated tensors within the node. As for our hierarchical implementation, there are no leader worker within a node, and each work will identically  process a subdivision of the tensor, details please refer to Section 4. Moreover, to the best of our knowledge, there is no collective communication primitive that can "all reduce" 1-bit data in DDP. If we use the standard summation *All-Reduce*, the data will be overflowed. Although we can use the standard *All-Gather* among workers to aggregate data, the communication efficiency is much inefficient. Therefore, we utilize *Scatter-Reduce*, *All-to-All* and some necessary computation operations between these communication primitives to construct *Hierarchical 1-bit All-Reduce*. Hence, we think the specifically-designed *Hierarchical 1-bit All-Reduce* has its unique contribution.
> > >
> > > **Q5.** BinSGM is not the first work to be consistently faster than DDP. Existing works like Bagua and HiPress already show significant speedup over DDP.
> > >
> > > **R5.** There may be some misunderstandings. We might not claim that BinSGDM showed speed superiority to DDP, we just claimed that running with DDP, BinSGDM is the first work to be consistently faster than the full-precision Adam and SGDM, compared to existing gradient compression optimizers. BinSGDM is an optimizer, and DDP, Bagua and HiPress are data parallel frameworks. Optimizers and data parallel frameworks are othogonal ,  so It is inappropriate to directly compare BinSGDM to DDP, Bagua or HiPress. Actually,  BinSGDM running with  Bagua or HiPress that is specifically designed for compressed optimizers may achieve better performance.
> > >
> > > *We hope that our response addresses your questions and concerns. If so, it would be great if you can raise your rating.*

---

> ### Author Response · Authors · 2022-12-09
> **Eagerly looking forward further post-rebuttal feedback with Reviewer C7p2 (the disccusion deadline is due in two days)**
>
> Dear Reviewer C7p2,
>
> We keenly thank you for your original constructive suggestions and comments.  As you suggested, we had implemented comparison experiments with 200Gbps InfiniBand, and comparison experiments for the uncompressed version of the proposed BinSGDM and Adam on extensive tasks. Also, we had clarified the problems you are concerned about -- the convergence rate of BinSGDM, the novelty of *Hierarchical 1-bit All-Reduce*, and the relationship between BinSGDM and Existing works like Bagua and HiPress.  For the details, please see our previous responses.  Since our paper is at the borderline,  it is great that you could reconsider your score if we have addressed your concerns. If you have any more questions, we really hope to have a further discussion with you.  Thank you!
>
> Sincerely,
>
> authors

---

### Official Review · Reviewer_kGZr · 2022-10-27

**Confidence:** 4
**Correctness:** 3
**Technical Novelty And Significance:** 2
**Empirical Novelty And Significance:** 3
**Recommendation:** 6

**Clarity, Quality, Novelty And Reproducibility:**

The algorithm is clearly presented (except the confusion between equation 4 and Alg. 1), and well-motivated, especially, the connection to the existing sign-SGD and Adam.
Moreover, all aspects of the experiments (HW, used software and libraries, architecture of the models, hyper-parameters, datasets, ...) are clearly stated with enough details.
However, as mentioned in the weaknesses, some parts of the paper are hard to understand or contain seemingly contradicting statements (e.g., first few sentences of Appendix B- Discussion). The writing of the paper needs some improvements.


**Strength And Weaknesses:**

**Strengths:**

The authors have considered all aspects of distributed DL in developing their algorithm, such as computational cost, communication, and aggregation. This paper is among the few works that have considered compressing model updates instead of the SGs, and the simulation results show that the training with Bin-SGDM is consistently faster (or at least as fast as) uncompressed optimizers (even with high-speed intra-node connections among multiple GPUs).

**Weaknesses:**

In some parts, the paper is hard to read and the intent of the authors of some sentences/paragraphs are not clear. Moreover, some notations are not defined, and the reader has to search the appendix for the definition of some of the concepts. Moreover, the theoretical analysis seems to be flawed at some points.

1. There is a disparity between algorithm 1 and BinSGDM equation (4). Algorithm 1, line 7, keeps the maximum of $b_t^{(i)}$ and uses it to compute the parameter update. The authors do not explain or motivate this seemingly critical point in distributed BinSGDM, and it is not clear whether equation 4 or Alg. 1 is used for simulations. Practically, since the magnitude of SGs are usually larger at the first few training epochs, using the maximum value ($b_t^{(i)}$) is similar to scaling the SGs with a large constant value to map them to [-1, 1]. In other words, after few epochs, Alg. 1 applies (almost) fixed scaling factors to the computed SGs before binarizing them.
2. Theorem 1, what are the constants $\rho$ and $I$? Also, $z_1$ should be defined in the theorem.
3. The analysis and proofs in the appendix either have some flaws or unclear:
- Equation 10, note that although $g_t=\frac{1}{n}\sum_i g_t^{(i)}$, but $|g_t| \neq \frac{1}{n}\sum_i |g_t^{(i)}|$. Hence, the RHS equation is incorrect. I am not sure how this affects the validity of the proofs.
- Equation 34, the reason for 3rd and 4th inequalities are not clear.


**Summary Of The Paper:**

A major bottleneck in large-scale distributed deep learning is the communication bottleneck. The computational cost of the majority of existing compression algorithms to reduce the communication cost is too high. The authors proposed a new distributed optimization algorithm, Binary SGD-Momentum which 1- compresses the updates of the model's parameters at each worker, instead of the gradients commonly done in most other distributed DL methods, and 2-stochastically quantizes the values to +1 or -1. To aggregate the binary quantized values, they devise hierarchical 1-bit All-Reduce to take advantage of the 1-bit quantized values, and inter-node/intra-node communications. Finally, they theoretically analyzed their algorithm and evaluated it performance using numerous experiments.


**Summary Of The Review:**

The paper has presented an interesting and somehow novel compression algorithm for the distributed DL.The authors have provided both theoretical analysis and to some extent complete simulation results.
I believe that the paper needs some minor improvements, such as double-checking the proofs of theorems and fixing some mistakes,

---

> ### Author Response · Authors · 2022-11-15
> **Response to kGZr  Part 1/2**
>
> First of all, we would like to sincerely thank you for your time and your valuable and careful comments. We are thankful for your recognition of the novelty of this manuscript and its potential impact on the distributed DL. We are also grateful to you for acknowledging the clear presentation and motivation of the algorithm.  We regret that there were some writing issues and typos, as we were in a hurry to meet the deadline and did not have time to double-check the original manuscript. We have addressed them carefully in the revised version. In the following,  we will explain your concerns point by point.
>
> **Q1.** The disparity between Equation (4) and  Line 7 in Algorithm 1.
>
> **R1.** In Equation (4), we sketch BinSGDM to introduce the motivation, and some details are omitted.  Line 7 in Algorithm 1 is  added to theoretically guarantee satisfying convergence, following the idea of [1] ( that is awarded as the best paper in ICLR2018 ). The main contribution of [1] is that it identifies a problem in the proof of convergence of the widely-used original Adam, erroneously assuming $\frac{\alpha_{t-1}}{\sqrt{(v_{t-1})_j}} - \frac{\alpha_t}{\sqrt{(v_t)_j}}\ge 0$ (where $\alpha_t$ is the learning rate, $(v_t)_j$ is the $j^{th}$ element of the second-order momentum) always holds, and [1] constructs a counterexample. [1] then proposes a new optimizer, referred to as GMSGrad,  which corrects the original Adam by adding $v_t = \max(v _{t-1}, \hat{v}_t) $ (the non-decreasing denominator technique). And we also adopt this technique and add Line 7 in BinSGDM, \emph{i.e.}, $b_t = \max(b _{t-1}, \hat{b}_t) $. Actually, all the Adam-type optimizers will suffer from this theoretical convergence problem if the non-decreasing denominator technique is not adopted.
>
> Thank you for pointing out this important issue that gradient may be overlarge at early iterations. We were not aware of this problem before.  Fortunately, in the implementation of BinSGDM, $\hat{b}_0$ and $b_0$ are initialized to $0$, and the updating rule is $\hat{b}_t = \beta \hat{b} _{t-1} + (1-\beta)g_t, b_t = \max(b _{t-1}, \hat{b}_t)$, , and we commonly set $\beta$ to a value that is very close to $1$. At the early stage,  large $g_t$ will be neutralized by small $1-\beta$ and $\hat{b}_0 =0$, so that $b_t$ will not become too large. Hence, the large gradient at first iterations may not make $b_t$ fixed in the rest iterations. In practice, when training  DNNs, the performance  for  an Adam-type optimizer  with the non-decreasing denominator technique above is little influenced by the potential large gradients at early iterations, which is also validated in the experiments in [1][2]. Also, the practical convergence for Adam with or without  $v_t = \max(v _{t-1}, \hat{v}_t) $ and BinSGDM  with or without $b_t = \max(b _{t-1}, \hat{b}_t) $ makes little difference. In other words, the non-decreasing denominator technique is more about theoretical significance. In our experiments, we did not perform the non-decreasing denominator technique for BinSGDM to save the time for communication. In the revised version, we have addressed this problem in a straightforward way. We set a threshold, and when the number of iterations is larger than the threshold, the non-decreasing denominator technique  will take effect. We have included this method and the explanation above in Algorithm 1 in the revised version.
>
> [1] Sashank J. Reddi, et. al., *On the Convergence of Adam and Beyond*, ICLR, 2018.
>
> [2] Liangchen Luo, et. al., *Adaptive Gradient methods with Dynamic bound of learning rate*, ICLR, 2019.
>
> **Q2.**  $b_t^{(i)} \succeq \rho I$ and the definition of $z$ in Theorem 1.
>
> **R2.** Thank you for pointing out this less rigorous statement. The original statement $b_t^{(i)} \succeq \rho I$ is confusing and a little problematic,  so we revise the statement as $(b_t^{(i)})_j \ge \rho >0 $, $\forall j \in [1,2,...,d] $, and we have also added the definition of $z_t$ in Theorem 1 as you suggested.
>
> **Q3.** The problem about the rightmost part of Equation (10).
>
> **R3.**  We are sorry for this typo. Equation 10 should be $b_t = \frac{1}{n}\sum_{i=1}^n b_t^{(i)}$ rather than $b_t= \frac{1}{n}\sum_{i=1}^n b_t^{(i)}=\beta b _{t-1} + (1-\beta)|g_t|$, and the rightmost part is not needed. We miswrote Equation (10) in reference to Equation (9) ,$m_t= \frac{1}{n}\sum _{i=1}^n m_t^{(i)} = \beta m _{t-1} + (1-\beta)g_t)$, and did not double check it. Notably, in our proof of Theorem 1, for $b_t$,   we just require $-1 \le (\frac{m_t}{b_t})_j \le 1, \frac{1}{(b _{t})_j} \le \frac{1}{(b _{t-1})_j}$, $\forall j \in [1,2,..., d]$,  and it is obvious that the definition in Equation 10 can meet this requirement. In other words, we did not use the erroneous part  $b_t =\beta b _{t-1} + (1-\beta)|g_t| $ in the proof, so this typo will not influence the validity of the proof.

---

> > ### Author Response · Authors · 2022-11-15
> > **Response to kGZr  Part 2/2**
> >
> > **Q4.** The reasons for the 3rd and the 4th inequalities in Equation (34).
> >
> > **R4.**  Thank you for checking so carefully. We re-derive the 3rd and 4th inequalities in Equation (34) here with more details. For the core of the 3rd inequality in Equation 34, we have $\sum_{t=1}^T\left\Vert\frac{\alpha_{t-1} }{b_{t-1}} - \frac{\alpha_{t} }{b_{t}}\right\Vert^2 \le \sum _{t=1}^T \max_j \left\vert \frac{\alpha _{t-1} }{(b _{t-1})_j} - \frac{\alpha _{t} }{(b _{t})_j} \right\vert \left\Vert\frac{\alpha _{t-1} }{b _{t-1}} - \frac{\alpha _{t} }{b _{t}}\right\Vert_1 \le \sum _{t=1}^T \max_j \frac{\alpha _{t-1} }{(b _{t-1})_j} \left\Vert \frac{\alpha _{t-1} }{b _{t-1}}- \frac{\alpha _{t} }{b _{t}}\right\Vert_1 \le \frac{\alpha_1}{\rho} \sum _{t=1}^T \left\Vert\frac{\alpha _{t-1} }{b _{t-1}} -\right\Vert_1 $ where the first inequality holds due to the fact $\Vert a \Vert^2 \le \max_j|(a)_j| \Vert a \Vert_1$, the second inequality holds owing to $\frac{\alpha _{t-1} }{(b _{t-1})_j} - \frac{\alpha _{t} }{(b _{t})_j} > 0$ and $\frac{\alpha _{t} }{(b _{t})_j}>0$, the third inequality holds resulting from the given conditions $(b_t)_j \ge \rho >0$ for any $j$ and $\alpha_t \le \alpha_1$. For the core of the 4th inequality in Equation 34, we have $\sum _{t=1}^T\left\Vert\frac{\alpha _{t-1} }{b _{t-1}} - \frac{\alpha _{t} }{b _{t}}\right\Vert_1 = \sum _{t=1}^T\left\Vert\frac{\alpha _{t-1} }{b _{t-1}} \right \Vert_1- \left \Vert \frac{\alpha _{t} }{b _{t}}\right\Vert_1= \Vert\frac{\alpha _{1} }{b _{1}}\Vert_1 - \Vert\frac{\alpha _{T} }{b _{T}}\Vert_1 \le \Vert\frac{\alpha _{1} }{b _{1}}\Vert_1 \le \frac{\alpha_1 d}{\rho}$ where the first equality holds due to  $\frac{\alpha _{t-1} }{(b _{t-1})_j} - \frac{\alpha _{t} }{(b _{t})_j} > 0$, the second equality holds owing to telescoping sum and ${\alpha _{0} } = {\alpha _{1} }$, $b_0=b_1$, and the last inequality holds resulting from $(b_1)_j \ge \rho >0$ for any $j$. We missed $\alpha_1$ in the derivation in the original manuscript that might make you confused.  We have added more details for Equation (34) in the updated version.
> >
> > **Q5.**  Seemingly contradicting statements in the first few sentences in Appendix B.
> >
> > **R5.**  We re-read these sentences carefully, but we do not find the problem. Please specify what the contradicting statements are.
> >
> > *We hope our responses addresses your questions and concerns. If so, it would be nice to raise your rating.*

---

### Author Response · Authors · 2022-12-06
**Sincerely expecting deep discussions with reviewers**

Dear reviewers,

We greatly thank you for your valuable time and reviewing.  We believe our work can contribute a lot to distributed DNN training. The final discussion deadline (12/12) is approaching, and we look forward to deeply discussing with you to see if our responses and revisions in the manuscripts had addressed your concerns. Currently, our paper is at the borderline, your kind reconsideration may directly influence the fate of the manuscript. Thank you again!

Sincerely,

authors

---

### Decision · Program_Chairs · 2023-01-20

**Decision:**

Reject

**Justification For Why Not Higher Score:**

Fatal issues with the theory and assumptions.

**Justification For Why Not Lower Score:**

N/A

**Metareview: Summary, Strengths And Weaknesses:**

Communication bottleneck is a major bottleneck in large-scale distributed deep learning. The authors propose a new distributed optimization algorithm, Binary SGD-Momentum, which compresses the updates of the model's parameters at each worker. To aggregate the binary quantized values, they devise a hierarchical 1-bit All-Reduce protocol to take advantage of the 1-bit quantized values and inter-node/intra-node communications. They theoretically analyze their algorithm and evaluate its performance numerically.

Weaknesses:

1. The authors claim: "All the existing communication-efficiency optimizers are built upon gradient compression. In contrast, to the best of our knowledge, we are the first to directly quantize the entire model update, which will streamline the quantization." The authors are not aware of an entire subfield of methods (worth tens to hundreds of papers) which quantize model updates. Examples of such methods: DIANA (the first method that achieved this), EF21 (does this for biased compressors such as top-k), MARINA (currently the theoretical SOTA in the nonconvex regime) - and many follow up works. These works show that compressing model updates is theoretically superior to compressing models. This is an important omission.

2. Assumption 4 on (deterministically) bounded stochastic gradients is highly problematic and is not justified.

3. The rate in Theorem 1 does not make sense. One of the reasons is: Theorem 1 assumes that the sequence $( b_t^{(i)} )_j$ is bounded below by some $\rho >0$ throughout the iterations. However, this is not proved, so it may not be true. Even if it is true, it is not clear how small $\rho$ can be. The bound in (6) depends inversely in $\rho$, and becomes vacuous for small values of $\rho$.

4. The writing seems hurried, with many grammatical and stylistic issues.

These are examples of several important issues with the paper; some of them being fatal. The reviewers noticed more issues than this. Some reviewers had a borderline positive view of the paper, but I did not see sufficient reasoning and explanation in these reviews that would be convincing.

The experimental part of the paper seems to be the key highlight.

I recommend the authors to remove the theoretical part and focus on the empirical results, or to make a realistic and critical assessment of the results and assumptions, so that the reader would not be mislead but enlightened.

Unfortunately, I cannot recommend the paper for acceptance in its current form.

**Summary Of Ac-Reviewer Meeting:**

There was no AC-reviewer meeting as I did not think one was needed. This is because I've read the paper, which helped me to form an opinion.

---

> ### Author Response · Authors · 2023-02-24
> **Responses to the decision**
>
> Although the final results cannot be changed, we would still like to clarify the "flaws" in the metareview comments.
>
> 1. It appears that there may be some misunderstandings regarding our contribution. We would like to emphasize that  we just claimed  *"we are the first to directly quantize the entire model update  **of an adaptive optimizer** "*, rather than  *"we are the first to directly quantize the entire model update"*, as mentioned in Contribution \#1 in the Introduction section of our paper. The well-known methods mentioned, such as DIANA, EF21, and MARINA, are the quantized surrogtes of GD or SGD, while our proposed method is a quantized surrogate of an adaptive optimizer. Nowadays, adaptive optimizers, including Adam and its variants, are dominant in training deep DNNs, such as Transformer-based networks. Although research on how to quantize adaptive optimizers is important for distributed training of DNNs, it was still unexplored before our work. Therefore, we believe that our work on how to quantize adaptive optimizers for distributed training of DNNs is a significant contribution to the field. Due to space limitations, we only cited some other classic quantization-SGD methods, such as 1-bit SGD, SignSGD, QSGD, and so on, rather than the methods you mentioned.
>
> 2. Assumption 4 actually describes the bounded gradient oracle, i.e., $\Vert \nabla f(x) \Vert \le G$, which is a common and necessary assumption in previous papers to theoretically demonstrate the convergence of the adaptive optimizer, as can be found in the literature on classic Adam, AMSGrad, AMSBound, Adan, and so on. More importantly, Assumption 4 can be satisfied when training DNNs in practice. Therefore, we maintain that the assumption is reasonable and necessary for the theoretical analysis of our work.
>
> 3. We add a small constant to $b_t$ in practice, i.e., $b_t = b_t + \epsilon$, to avoid a zero denominator for numerical stability. This ensures that the actual update rule is $x_{t+1} = x_t -\alpha {\mathcal Q}(\frac{m_t}{b_t+\epsilon})$, where $m_t = \beta g_t + (1-\beta)g_t$ and $m_t = \beta g_t + (1-\beta)|g_t|$. This update guarantees that the assumption that the sequence $(b_t)_j$ is bounded below by some $\rho > 0$ throughout the iterations can always be satisfied. Notably, the explanation above is outlined in Footnote #3 of our paper. Additionally, in all adaptive optimizers, it is a necessary step to add a small constant to the denominator of the update in the update rule to avoid a zero denominator for numerical stability.